

# How do changes along the risk chain affect flood risk?

Ayse Duha Metin[1], Nguyen Viet Dung[1], Kai Schröter[1], Björn Guse[1,3], Heiko Apel[1], Heidi Kreibich[1], Sergiy Vorogushyn[1], Bruno Merz[1,2]

[1]GFZ German Research Centre for Geosciences, Section Hydrology, 14473 Potsdam, Germany
[2]Institute of Earth and Environmental Science, University of Potsdam, 14476 Potsdam, Germany
[3]Department of Hydrology and Water Resources Management, Kiel University, Germany

*Correspondence to*: Ayse Duha Metin (admetin@gfz-potsdam.com)

**Abstract.** Flood risk is impacted by a range of physical and socio-economic processes. Hence, the quantification of flood risk ideally considers the complete flood risk chain, from atmospheric processes through catchment and river system processes to damage mechanisms in the affected areas. Although it is generally accepted that a multitude of changes along the risk chain can occur and impact flood risk, there is a lack of knowledge how and to what extent changes in influencing factors propagate through the chain and finally affect flood risk. To fill this gap, we present a comprehensive sensitivity analysis which considers changes in all risk components, i.e. changes in climate, catchment, river system, land use, assets and vulnerability. The application of this framework to the mesoscale Mulde catchment in Germany shows that flood risk can vary dramatically as consequence of plausible change scenarios. It further reveals that components that have not received much attention, such as changes in dike systems or in vulnerability, may outweigh changes in often investigated components, such as climate. Although the specific results are conditional on the case study area and the selected assumptions, they emphasise the need for a broader consideration of potential drivers of change in a comprehensive way. Hence, our approach contributes to a better understanding of how the different risk components influences the overall flood risk.

## 1. Introduction

Globally, floods affect more people than any other natural hazard, and the global average annual flood loss has been estimated to amount to more than US\$ 100 billion (UNISDR, 2015). Flood risk is defined as the likelihood of losses and depends on three factors: hazard, exposure and vulnerability (IPCC, 2012; UNISDR, 2013). Hazard is related to the physical processes with the potential to cause harm ranging from atmospheric via catchment processes to river routing, whereas exposure refers to the elements-at-risk of flooding. Vulnerability is defined as the susceptibility of the elements-at-risk to be adversely affected. Typically, exposure is quantified as the number of people and the assets in flood-prone areas, and vulnerability is represented as the damage ratio, i.e. the degree to which elements-at-risk are damaged given hazard impacts. Consequently, flood risk assessments ideally need to consider the entire flood risk chain from the atmospheric processes, through the catchment and river system processes to the damage mechanisms in the affected areas.

It is now well acknowledged that flood risk can change substantially in time, since all three risk factors are dynamic (Kreibich et al., 2017). The causes of these changes are manifold; they range from human-induced climate change and natural climate variability on decadal or centennial time scales to changes in vulnerability that may act on much shorter time scales (Merz et al., 2010). The spatial and temporal interdependencies between hazard, exposure



and vulnerability and interactions within these risk chain compartments should be considered in flood risk assessment (Merz et al., 2014; Vorogushyn et al., 2017).

In their study of paired flood events, Kreibich et al. (2017) looked into consecutive flood events that occurred in the same region and attempted to understand what drove the changes in the observed impact. Their collection of case studies revealed the essential role of vulnerability reduction on losses, for instance, via improved risk
awareness, preparedness and organizational emergency management. On the other hand, they emphasized that different risk drivers act simultaneously, for instance structural measures can be complemented by non-structural measures.

Another approach to understand changes in flood risk is loss normalization using observed damage data (Visser et al., 2014). Time series of flood damages show usually increasing trends. To separate the effect of socio-economic
development, the original loss time series are corrected for growth in population and wealth, and for inflation. For example, Barredo (2009) normalized losses of large river floods aggregated at the scale of 31 European countries between 1970 and 2006. Since the normalization removed the increasing trend in the original loss values, this study suggested that socio-economic development was the dominant driver of increasing flood damage in Europe. Similar conclusions have been drawn from other loss normalization studies for weather-related hazards (IPCC,
2012; Neumayer and Barthel, 2011; Bouwer, 2011; Visser et al., 2014).

Other data-based studies attempted to understand the influence of single drivers. For instance, Bubeck et al. (2012) surveyed 752 households along the Rhine and found that the implementation of private mitigation measures developed gradually over time with severe floods leading to a stepwise increase in mitigation. They concluded that an improved preparedness triggered by a severe flood in 1993 led to substantial damage reduction during a second
flood with similar hazard characteristics in 1995. A survey of 1200 households affected by the Elbe flood in 2002 in Germany suggested that private precautionary measures reduced the damage to the building and contents in the order of 50 % for the most effective measures, i.e. flood adapted use and adapted interior fitting (Kreibich et al., 2005).

Although data-based approaches have helped to better understand flood risk changes, it is hard to conceive how
the causes of flood risk changes and their relative contributions could be deciphered from empirical data only. A major problem is the superposition of several drivers of risk changes. It is easily conceivable that adaptation measures, such as improved early warning systems, strengthened flood protection or better private precaution, have masked the effect of climate change (Handmer et al., 2012; Di Baldassarre et al., 2015; Jongman et al., 2015; Mechler and Bouwer, 2015). Hence, conclusions from normalization studies, such as no evidence for the effect of
human-induced climate change on the loss trend (e.g. Barredo, 2009), need to be taken with care. Another limitation of data-based approaches results from the lack of reliable loss data. Loss data are often not available, or are available only for standard economic sectors in developed countries, and large uncertainties reside in reported or reconstructed loss records (Handmer et al., 2012; Merz et al., 2010; Wirtz et al., 2014).

Simulation-based approaches offer the advantage that the contributions of different drivers can be estimated via
scenario runs. Table 1 compiles simulation-based studies that investigated past or future changes in river flood risk. The various studies that addressed changes in flood hazard only, for instance as consequence of climate and land use change, are not included. This selection of studies results from a comprehensive literature search using the following search terms in the ISI Web of Knowledge database: *flood risk, change, damage, climate* and



*socioeconomic scenarios* in October 2017. The identified articles were checked for forward and backward citations. We would like to point out that studies focussing on the uncertainties in estimation of hazard, exposure, vulnerability and their effect on risk estimates were not in the focus of this review.

Table 1 shows that all studies addressed climate change. Other changes in flood hazard have not been investigated with the exception of land subsidence by Budiyono et al. (2016). Almost all studies look at changes in exposure, most often in terms of land use change. Changes in asset values are also addressed frequently. In terms of risk indicators, the majority of studies is limited to EAD (Expected Annual Damage).

There is no unanimous conclusion across these simulation-based studies. The results highly depend on the case study and the drivers and scenarios selected. Yet, 5 out of 13 studies concluded that climate change was the dominant driver leading to an increase in flood risk. The other studies indicated different drivers and combinations as more dominant. (For a detailed assessment of these studies see the supplementary material.)

Although there is a wealth of studies on how and why flood hazard has changed in the past and might change in the future (IPCC, 2012), studies on changes in flood risk are scarce. Data-based approaches are strongly limited due to data availability and methodological problems. Simulation-based studies on changes in flood risk have been limited to climate and land use change and have primarily focussed on future scenarios rather than understanding past changes. Other drivers of risk, such as flood protection measures, have been neglected. This gap is particularly severe in terms of the effects of changes in vulnerability (Merz et al., 2014; Mechler and Bouwer, 2015). Our systematic literature search did not result in a single simulation-based study which included changes in vulnerability. We can conclude that knowledge about the underlying processes and their contribution to changes in flood risk is still scarce (UNISDR, 2015; Kreibich et al., 2017), and there is a lack of comprehensive studies that take into account the whole spectrum of drivers.

Our study is a contribution to fill this research gap. It analyses how different drivers, including all three components of risk, affect flood risk. Changes in flood risk are evaluated for the catchment scale and two typical up- and downstream sub-basins and for summer and winter seasons. We quantify the sensitivity of flood risk to changes along the flood risk chain, considering all components of the chain. This includes changes in the atmosphere, catchment, river system and affected floodplain areas. Specifically, we consider climate change, implementation of reservoirs in the catchment, flood protection along the rivers, land use change, change in asset values and changes in the vulnerability of flood-affected objects. For each of the six factors, two scenarios with increasing and decreasing change with symmetric deviation from a baseline scenario are derived. Hence, the sensitivity analysis consists of 729 ($3^6$) scenarios.

This sensitivity analysis is combined with the 'Derived Flood Risk Analysis (DFRA)' proposed by Falter et al. (2015). DFRA consists of an end-to-end flood risk assessment based on continuous simulation. A model chain representing the catchment, river network and damage processes is driven by a multi-site stochastic weather generator. DFRA is an extension of the 'Derived Flood Frequency Analysis' based on continuous simulation which has found increasing attention recently (e.g. Haberlandt and Radtke, 2014). A major advantage of DFRA is that all processes, from the flood-triggering precipitation to the damage, are simulated in a spatially consistent way, respecting the spatial dependence of the different processes. Another advantage is the derivation of flood risk directly from the damage time series, generated by the model chain, instead of the discharge time series.





The sensitivity analysis is performed for the Mulde catchment in Germany which has been severely hit by flooding in 2002 and 2013. We use the model chain implemented and calibrated by Falter et al. (2015) for the Mulde catchment. 4000 years of spatial weather fields at daily resolution are generated and used to force the model chain, resulting in daily and spatially explicit fields of streamflow, inundation and damage throughout the catchment. From these data sets, the risk curve (or loss-probability curve) and EAD are calculated. Introducing the change scenarios for the six factors leads to 729 damage time series of length 4,000 years which again are used to calculate the flood risk.

The paper is structured in six sections. Section 2 describes the study area. Section 3 introduces the simulation model chain and the approach used in the sensitivity analysis including the change scenarios. Section 4 presents the results of the sensitivity analysis including sub-basin and sub-annual variations. Sections 5 and 6 provide discussions and conclusions.

## 2. Study area

Our study area, the Mulde catchment (7115 km²), is a sub-basin of the Elbe River in Germany which is one of the largest rivers in central Europe. The Mulde River drains the northern part of the Ore Mountains. The Mulde and its major tributaries have a length of around 380 km. The catchment elevation varies between 52 m and 1213 m above sea level. Approximately 10 % of the catchment area is covered by urban structures. Anhalt-Bitterfeld, located downstream in the Mulde catchment, and Zwickau, located upstream, have been selected as two districts for more detailed analyses (Figure 1). The annual precipitation ranges from 500 mm to 1100 mm. Although the majority of floods in the Mulde catchment occurs in winter, extreme floods tend to occur in summer due to widespread and intensive precipitation. Reservoirs in the Mulde catchment (14 of them have storage capacity greater than 1 million m³) are generally used for drinking water supply, but also they have storage capacity for flood protection (Schädler et al., 2012).

The most extreme floods during the last decades in Germany were observed in August 2002 and June 2013 (Schröter et al., 2015). While the 2002 flood has been the most expensive disaster for Germany to date, the 2013 event has been the most severe flood in hydrological terms in the last six decades. Both floods had also severe impacts in the Mulde catchment. 115 and 24 dike failures were observed in the Mulde catchment in 2002 and 2013, respectively (Thieken et al., 2016). Historical documents, going back to the 9[th] century, show that the Mulde catchment has been hit by large floods associated with high damages (Petrow et al., 2007). The repeated occurrence of extreme flooding associated with high damages is the primary reason for selecting it as study area.




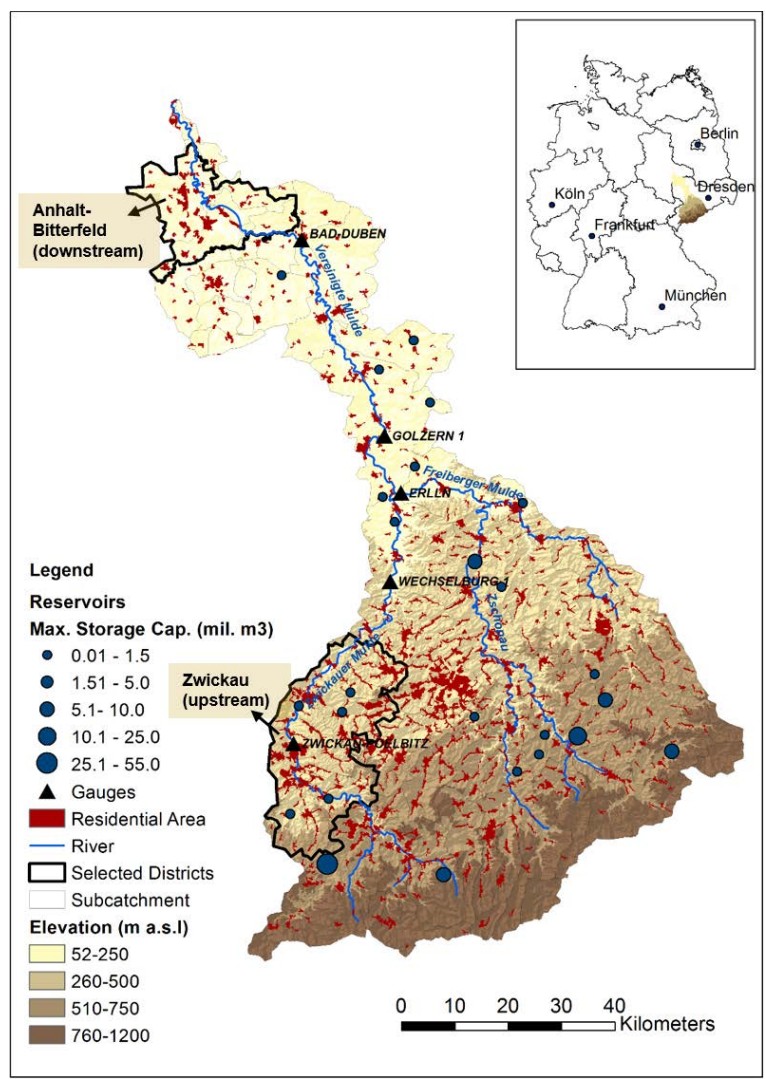

**Fig. 1. Study area Mulde catchment, including main tributaries, reservoirs and river gauges. The inset shows the location of the catchment within Germany.**

**3. Methods**

**3.1. Flood risk simulation model chain**

To simulate the complete flood risk chain, the Regional Flood Model (RFM) is used. RFM consists of a weather generator, rainfall-runoff model, 1D channel routing model, 2D hinterland inundation model and flood loss estimation model for residential buildings. The results of one model are used as input for the next model. Fig. 2

shows the model chain and gives the most important information on the input data and the characteristics of the different modules. Details about the model chain are given in Falter et al. (2015).

The model setup follows the concept of derived flood risk analysis based on continuous simulation proposed by Falter et al. (2015). A weather generator provides spatially consistent meteorological fields which propagate



through the entire model chain. In our study, the chain is run on a daily time step for 40 realizations of 100 years
resulting in a total time series of 4000 years. Risk estimates are then derived directly from the time series of damage
generated by the model chain.

A derived flood risk analysis based on continuous simulation has a number of advantages compared to event-based
flood risk estimates. For instance, due to the continuous simulation the antecedent catchment conditions are
implicitly considered in the flood generation, and the approach provides the complete flood hydrograph. Since all
models within the chain are spatially explicit, the approach provides spatially consistent flood events including the
river-floodplain and damage processes. Hence, the spatial dependence between flood damages at different
locations in the catchment is taken into account. A further advantage is that risk is estimated using the space-time
fields of damage. Hence, this approach follows the definition of risk, where risk is understood as the probability
of exceeding a given damage. In contrast, traditional flood risk analyses use the probability of discharge as proxy
for the probability of damage. For a comprehensive discussion see Falter et al. (2015).

Note that our model setup is the same as in Falter et al. (2015). The only difference is that we consider reservoirs
in the rainfall-runoff module. The different modules along the risk model chain are described in the following.





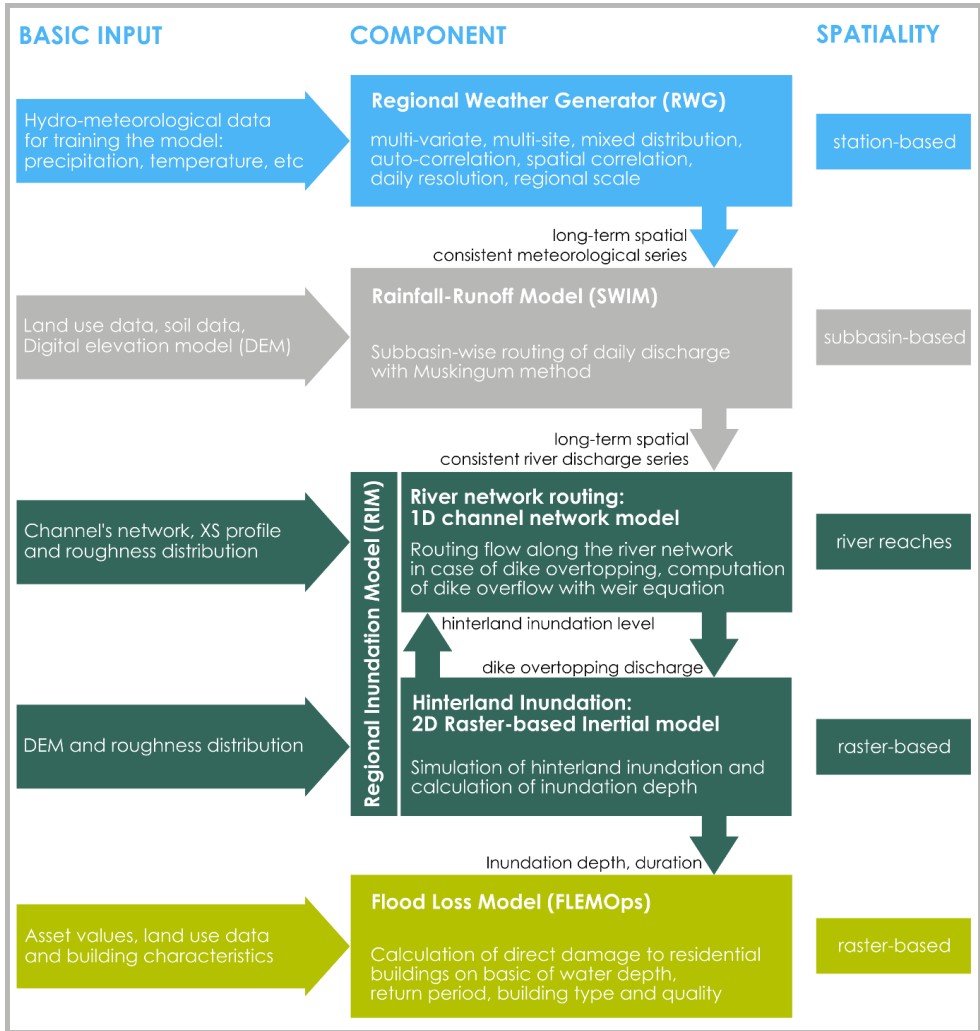

**Fig. 2. Flood risk model chain: Regional Flood Model (RFM)**


### 3.1.1. Regional weather generator RWG

The meteorological input is obtained from the multi-site, multi-variate weather generator RWG (Regional Weather Generator) proposed by Hundecha et al. (2009) and further developed by Hundecha and Merz (2012). This model is designed to generate synthetic weather at the regional scale, i.e. several 10,000 to 100,000 km$^2$. It creates daily

time series of climatic variables at multiple sites in two steps: generation of daily precipitation series through a multivariate-autoregressive model (which uses a mixed Gamma and Generalized Pareto distribution) and generation of daily maximum, minimum and mean temperature and solar radiation using Gaussian distribution. Both temperature and solar radiation depend on the state of precipitation. The weather generator is parameterized on a monthly basis.





The weather generator is set up for the whole of Germany, including the upstream areas of the Elbe, Danube and Rhine catchments outside of Germany. It is used to generate long synthetic meteorological data considering daily climate observations for the period from 1951 to 2003 at 528 climate stations.

All the single-site input parameters (six parameters of the mixed Gamma-Pareto distribution for non-zero precipitation and two parameters of the Gaussian distribution for the other variables) have been estimated for each
of 528 stations of the dataset and for each of 12 months separately. The RWG has been successfully tested and validated for the reproduction of daily and longer term statistics of the six climatic variables at individual sites and the reproduction of the temporal and spatial pattern observed in the dataset. The validation results illustrate that the RWG is capable of generating long-term, synthetic meteorological fields, capturing well both regular and extreme events. The detailed description of the implementation of the RWG would be extensive. Hence, for the
sake of simplicity and balance of the paper structure, it will not be elaborated here. The readers are referred to Falter et al. (2015) for more details.

### 3.1.2. Rainfall-runoff model SWIM

The semi-distributed hydrological model SWIM (Soil and Water Integrated Model, Krysanova et al., 1998) simulates the hydrological cycle on a daily basis. SWIM uses three levels of spatial disaggregation: the river basin
is divided into sub-basins which are further subdivided into hydrotopes. Water fluxes are computed at the hydrotope level, then aggregated on the sub-basin level. SWIM routes total runoff from sub-basin to sub-basin using the Muskingum routing method.

In this study, the Mulde catchment was divided into 77 sub-catchments based on Shuttle Radar Topography Mission digital elevation maps provided by the Federal Agency for Cartography and Geodesy in Germany (BKG).
Hydrotopes were formed using soil and land use data from the soil map of Germany (BÜK 1000 N2.3) from Bundesanstalt für Geowissenschaften und Rohstoffe, the European Soil Database map from the European Commission's Land Management and Natural Hazards unit, and the CORINE (COoRdinated INformation on the Environment) land cover map.

To be able to assess the sensitivity of flood risk to the implementation of reservoirs, we added a reservoir
component in SWIM. The specific operational strategy for each reservoir depends on a number of considerations. For example, after the disastrous flood in 2002, the storage reserved for flood retention has been increased at the expense of other purposes such as water supply for some reservoirs in Germany. The operational rules for reservoirs are expected to vary in time and from reservoir to reservoir based on local considerations. Further, it may be difficult to reconstruct them for reservoirs which have been in operation for decades. In this SWIM version,
a simplified routine was integrated for simulating the retention effect of reservoirs automatically. Each modelled reservoir is linked to the sub-basin in which it is located. When the flow at the sub-basin node exceeds the 100-year discharge ($HQ_{100}$), the streamflow beyond this threshold is stored in the reservoir, i.e. the hydrograph is cut at $HQ_{100}$, as long as the required storage volume is available. When the flow falls below the threshold value of $HQ_{100}$, the reservoir starts releasing water so that the flow maintains the level of $HQ_{100}$ as long as the active volume
allows. In total, 25 reservoirs (Fig. 1) within the Mulde catchment are integrated in the SWIM model setup. The necessary information for reservoirs was adapted from Sächsisches Landesamt für Umwelt und Geologie (2002).




The new SWIM model setup with reservoirs needed to be re-calibrated and re-validated using the identical dataset, global optimization algorithm (SCE-UA, Duan et al., 1992) and objective function mNSE (based on modified Nash-Sutcliffe efficiency measure giving more emphasis on higher flow) mentioned in Falter et al. (2015). The

calibration and validation periods remain the same as well (calibration: from 1-Janunary-1981 to 31-December-1989; validation: from 1-Janunary-1951 to 31-December-2003 excluding the calibration period). The calibration and validation results illustrate obvious improvement in this new model setup compared to the version used in Falter et al. (2015). At the upstream station Lichtenwalde, Nash-Sutcliffe values of 0.81 (calibration) and 0.83 (validation) are achieved for the new setup against 0.77 and 0.81 for the old one. At the downstream Mulde station

Bad Düben, the corresponding values are 0.89 and 0.86 against 0.89 and 0.83. More importantly, with the new setup, the SWIM model seems to be able to represent the cut-off process more accurately. With the new setup the modelled peak flow of the August 2002 flood fits well to the observed peak flow (Figure 3).

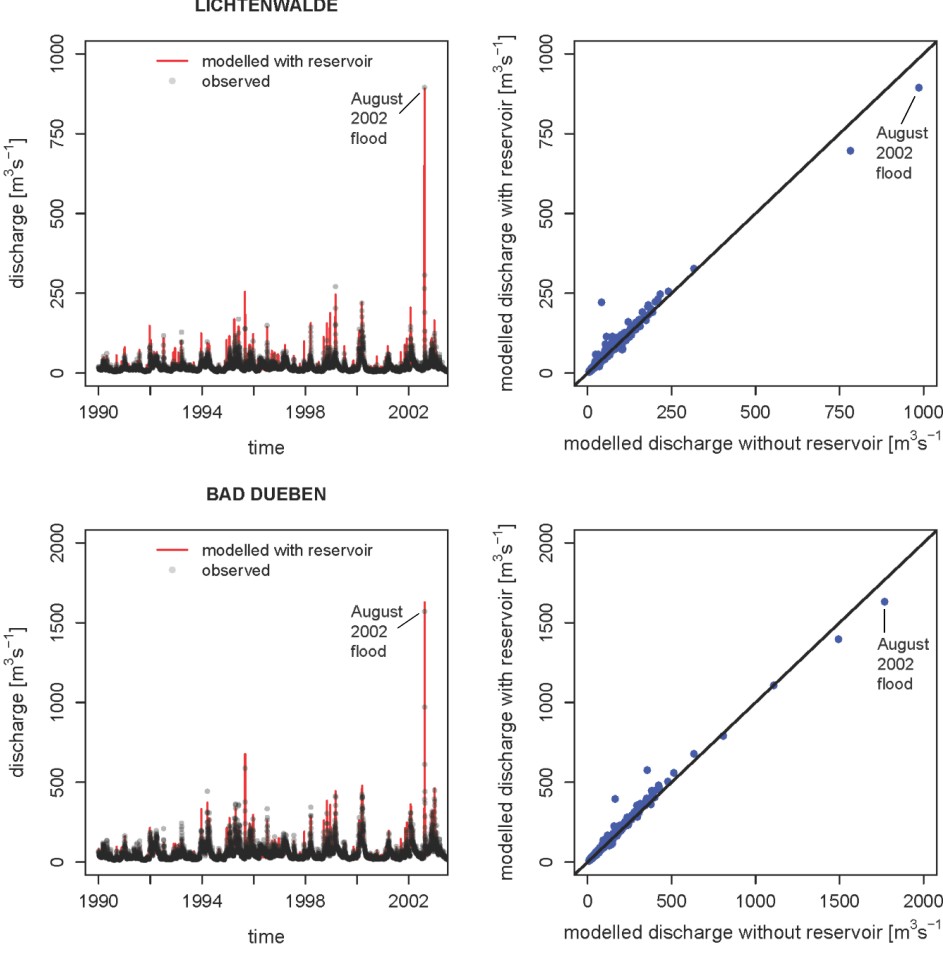



**Figure 3. Model performance of SWIM at selected gauging stations.**

**3.1.3. Regional inundation model RIM**

With the hydrological routing SWIM calculates wave propagation without explicit consideration of the river channel geometry. However, to predict dike overtopping and simulation of hinterland inundation, water level information along the river network is needed which is provided by the Regional Inundation Model (RIM). It consists of a 1D hydrodynamic channel routing model for the domain between river dikes and a 2D hydrodynamic

inundation model for the dike hinterland. Both models are coupled, i.e. the 1D model gives the overtopping flow as a boundary condition to the 2D model, and the hinterland water levels computed by the 2D model are used as boundary condition for the 1D model. The channel routing model solves the 1D diffusive wave equation using an explicit finite difference solution scheme and it simulates only the flood flows exceeding the bankfull discharge. To this end, the river cross-section geometry was simplified including the overbank river geometry and the

elevation of flood protection dikes. Whenever the water level reaches the dike crest level, overtopping flow into the hinterland is calculated using the broad-crested weir equation. Hinterland inundation processes are simulated with a 2D raster-based model based on the inertia implementation of Bates et al. (2010). The 2D inundation model was implemented in CUDA Fortran on Graphical Processor Units to increase the computational speed.

River cross-section profiles, dike heights and locations, and Manning's roughness values are necessary for setting

up the 1D model. The main data source for the geometric characteristics is the 10 m resolution digital elevation model (DEM) supplied by the Federal Agency for Cartography and Geodesy in Germany (BKG). Additionally, information on channel width and dike location was obtained from the digital basic landscape model (Base DLM) provided by BKG. The river profiles were manually extracted perpendicular to the flow direction with about 500 m spacing. Since the resolution of DEM 10 tends to provide too low dike heights, the minimum height was assumed

at 1.8 m. The Manning's coefficient of n=0.03 was adopted constant over the entire river network. The 2D raster-based model uses a 100 m resampled computational grid from DEM10, which was found an acceptable compromise for representation of inundation characteristics and computation time (Falter et al., 2013).

Falter et al. (2015) validated the 1D hydrodynamic model at five gauging stations (Fig.1) in the Mulde catchment with observed data over the period 1951-2003. Although there was a tendency to underestimate the number of

observed peak flows exceeding the bankfull depth, the general performance was acceptable. Validation of hinterland inundation is harder due to the lack of information about inundation depth and extent. In our study area, observed inundation is only available for the extreme flood of August 2002, provided by the German Aerospace Center (DLR). While inundation areas are simulated well for the eastern tributary Freiberger Mulde, only around 50% of the flood extent is correctly simulated for the entire catchment due to neglected dike breaches in the model

chain. Although there is an underestimation of inundation, the model gives a reasonable estimate of inundation extent and depth for large-scale assessments. Details can be found in Falter et al. (2015).

**3.1.4. Flood Loss Estimation Model FLEMOps**

The Flood Loss Estimation MOdel for the private sector (FLEMOps) is used to calculate direct economic damage to residential buildings for each inundation event using the maximum water level information provided by RIM.

The base version of FLEMOps uses five inundation depth classes, three building types, two building quality classes, three water contamination classes and three private precaution classes as inputs (Thieken et al., 2008). The



advanced version additionally considers the return period of the inundation at the flooded buildings as damage-influencing factor (Elmer et al., 2010, 2012). FLEMOps provides the damage ratio, i.e. the relative damage. The monetary damage is calculated by multiplying the damage ratio with the asset values of the exposed elements.

FLEMOps uses spatially detailed information about asset values, building types and building quality. Asset values of the regional stock of residential buildings were characterized considering standard construction costs (BMVBW, 2005). These asset values were spatially distributed according to the CORINE land cover classes 111 (continuous urban fabric) and 112 (discontinuous urban fabric). Municipal-scale information on building quality was provided by Infas Geodaten GmbH (2009). The flooding impact is characterised by inundation depth and return period of

peak flows. The latter is calculated at the SWIM sub-basin level by fitting a generalized extreme value distribution to the annual maximum discharge series obtained from 4000 years of continuous SWIM simulation.

The flood loss estimation was evaluated by Falter et al. (2015) for the 19 affected communities in the State of Saxony in Germany during flood event of August 2002. The sum of damages for all communities was officially reported as €240 million, and it was calculated as €67 million from the model chain. The simulated affected

residential areas match about 30% of the observed affected residential areas. This underestimation may be explained by uncertainty in asset values and their spatial distribution and uncertainty in the damage model. For details we refer to Falter et al. (2015).

### 3.2. Sensitivity analysis

### 3.2.1. Outline of the sensitivity analysis

We investigate the sensitivity of risk to changes in the flood risk chain components. To represent the entire flood risk chain, we analyse the effects of changes in the following six components: atmosphere (A), catchment (C), river system (R), exposure related to land use (EL), exposure related to asset values (EA), and vulnerability (V).

The most comprehensive approach for understanding model sensitivity is global sensitivity analysis where regression methods, screening-based, variance-based and meta-modelling approaches are widely used (Pianosi et

al., 2016; Song et al., 2015; van Griensven et al., 2006). Global sensitivity analysis evaluates the effects of all input parameters and their combinations on the output based on a large number of model runs. However, this approach cannot be combined with the derived flood risk analysis based on continuous simulation in our case study due to the massive computational time that would be required. Therefore, we use a much less demanding approach, the logic tree approach, to identify the contribution of each component to changes in flood risk and to understand

interaction effects by analysing all possible combinations.

For each component, we limit the sensitivity analysis to three scenarios, a baseline scenario and two symmetric change scenarios. The baseline scenario represents the current state. For example, the baseline scenario of the catchment component is represented by a model version calibrated for a recent time period and including the current implementation of reservoirs in the catchment. The specific time periods and assumptions for the baseline

scenarios are given in sections 3.1.1 to 3.1.4 where the implementation, calibration and validation of the different modules for the current situation are described. The change scenarios represent plausible deviations from the baseline. This setup leads to 729 ($3^6$) scenarios. The combinations of six components is shown in Figure 4.





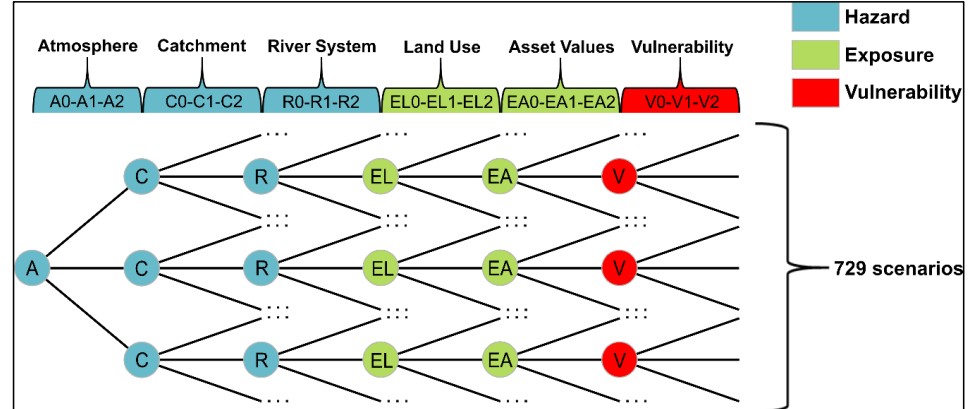

**Figure 4. Logic tree scheme for six components (atmosphere, catchment, river system, land use, asset values and vulnerability). For each component, there are one baseline (denoted by 1) and two symmetric change scenarios (denoted by 0 and 2).**

The variables that are changed for each component and their values for the baseline and change scenarios are described in the following sections and summarized in Table 2. It has to be noted that for a given component different types of changes would be possible. We have focussed our analysis on those types of changes that we consider most important for flooding in our study region. For example, changes in catchment hydrology are represented by changes in reservoir storage. Other changes, such as changes in agricultural practice possibly leading to changes in infiltration behaviour and runoff coefficients, are not considered. Further, the amount of change assumed for each component reflects another subjective choice. Finally, it should be noted that the change scenarios do not necessarily change the flood risk in the same direction. For example, scenario 2 of the catchment component represents increased flood retention capacity and, hence, reduced flood risk. On the other hand, scenario 2 of the vulnerability component assumes lower precaution compared to the baseline scenario, and hence, higher flood risk.

Each of the 729 scenarios consists of a continuous, spatially distributed simulation of the entire risk chain for 4000 years. From these resulting space-time fields of damage two risk indicators are analysed, namely the risk curve and the expected annual damage (EAD). The risk curve is obtained by plotting losses against their probability of occurrence. EAD is calculated by integrating over the risk curve. In this paper, we provide the results in aggregated form for the complete Mulde catchment, although the spatially explicit modelling setup allows deriving the sensitivity for each sub-catchment.

### 3.2.2. Change in climate

For the baseline scenario, the weather generator is calibrated using observation data from 1951 to 2003. We defined two plausible change scenarios considering seasonally different changes in precipitation and temperature. To apply these changes to the precipitation and temperature time series of the baseline scenario, we used the delta change method. For precipitation, the baseline time series of 4000 years of daily precipitation was multiplied by a change factor. For temperature, the change factor was added to the daily temperature time series of the baseline scenario. The change factors were derived from observed changes in mean seasonal precipitation and temperature across



Germany and are roughly representative for the past 50 years (Umweltbundesamt 2017a; 2017b). Scenario A2
represents a warmer climate and A0 a colder climate.

### 3.2.3. Change in catchment hydrology

Flood generation may be affected by a variety of mechanisms. Examples are land use changes, such as conversion
of agricultural areas into settlements or changes in infiltration behavior due to soil compaction as consequence of
more heavy machinery. We limit our analysis to changes in flood retention storage in reservoirs, which we consider
as the most important influence for the catchment component. Flood control by reservoirs is one of the dominant
flood risk management strategies in Germany. In upstream sub-basins of the Mulde catchment, flood retention
capacity of around 106 million $m^3$ has been implemented from 1825 to 2001 by constructing 25 reservoirs.

The baseline scenario C1 considers these 25 reservoirs. They were integrated into SWIM at their locations shown
in Figure 1. As change scenarios, we consider the catchment without reservoirs (scenario C0) and with double
storage capacity (scenario C2), respectively. In the latter case, we doubled the storage volume for each of the 25
reservoirs at the respective sub-basin.

### 3.2.4. Changes in the river system

For the river system, we focus on the effects of dikes on flood risk because dikes are the most extensively used
flood protection measure along rivers in Germany. The baseline scenario R1 represents the current situation with
the existing dikes.

To create change scenarios, we needed to define reasonable changes in dike height. The current height was
decreased (scenario R0) and increased (scenario R2) by 0.5 m, respectively. This increment is based on studies
about potential dike heightening in the Netherlands. Zwaneveld and Verweij (2014) considered 0.6 m dike
heightening, and Hoekstra and Kok (2008) compared two dike heightening strategies and for the better performing
approach, they assumed dike heightening in the range of 0.48 m to 0.71 m.

### 3.2.5. Land use change

Since the flood risk model chain used in this study considers only damage to private households, we limit the effect
of land use change to residential areas. The baseline scenario (EL1) considers the CORINE land cover classes 111
(continuous urban fabric) and 112 (discontinuous urban fabric) for the year 2012. The change scenario EL2 is
based on the increase in area of these two classes from 672 to 784 $km^2$ between 1990 and 2012 where the change
area was added to baseline scenario. To obtain the symmetric change scenario EL0, the same change in area (112
$km^2$) was subtracted from the situation in 2012. Pixels (100 x 100 $m^2$) of the classes 111 and 112 were assigned to
non-residential land cover classes (i.e. agricultural areas and semi-natural areas).

### 3.2.6. Change in asset values

For the baseline scenario (EA1), the building values from Kleist et al. (2006) for the year 2000 were converted to
2012 to be consistent with the baseline land use map. This conversion was based on the building price index (BPI)
which represents the growth in construction prices compared to a reference year for Germany (Baupreisindex-BPI,
DESTATIS, 2012). In agreement with the change scenarios for land use, we generated the change scenarios for
asset values by scaling the baseline scenario with the relative change in BPI between 1990 and 2012. Hence, the



change scenario EA2 represents a situation with a 34 % increase in asset values, and EA0 represents a 34 %
decrease compared to EA1.

### 3.2.6. Change in vulnerability

Vulnerability of private households is influenced by a variety of dimensions such as social, economic and
institutional, and it is challenging to quantify the relation between these dimensions and the damage ratio (Merz

et al., 2010). Therefore, in the present study, we focus on the economic dimension of vulnerability. To represent
changes in vulnerability, we use FLEMOps which was derived from comprehensive surveys of flood damage in
Germany (Thieken et al., 2008, Elmer et al., 2010). These surveys show that, besides flood and building
characteristics, contamination and precaution are significant factors in determining the damage. Since
contamination is in many cases imposed externally on households, for example by contamination through sewage

water, we focus our analysis on the effects of precaution.

The three vulnerability scenarios are defined by scaling the relative damage according to the level of precaution at
the household level. For medium contamination, the scaling factors are 1.20 and 0.71 for 'no precautionary
measures' and 'very good precautionary measures', respectively (Büchele et al., 2006). Hence, the change scenario
V2 with a scaling factor of 1.20 represents a situation without precautionary measures, and V0 a situation with

very good precaution (scaling factor 0.71). To obtain symmetrical changes, the scaling factor of the baseline
scenario V1 is set to 0.95.

### 4. Results

### 4.1. Sensitivity of flood risk at the catchment scale

The impact of each component on flood risk is illustrated in Figure 5 in terms of EAD, aggregated to the whole

Mulde catchment. Changes in each risk component are represented by three box plots, whereas each box plot is
derived from 243 scenarios for the change scenario 0, 1 and 2 of that risk component.

One of the most striking results is observed for the change in the river system. The median values for different
dike heights are €1.2 million, €0.8 million and €0.3 million for scenarios 0, 1 and 2, respectively. Hence, there is
a very strong reduction in EAD with dike heightening. The maximum EAD value for the high-dike scenario is

€1.1 million which is very low compared to the EAD values obtained across all scenarios. Another remarkable
result is the rather small increase in the median values for changes in the atmosphere (A) from scenarios 0 to 2
(from €0.6 million to €0.8 million), although climate change is generally addressed as the most influential
component. For the catchment (C) component, the median value for scenarios without storage capacity (C0) is €1
million, while it is around €0.6 million for scenarios with both baseline storage capacity and double storage

capacity. This non-symmetry in the effects of the catchment component is explained by the specific
implementation of the reservoir capacity: Implementing a capacity of 106 million m$^3$ reduces the EAD
significantly, but doubling this reservoir capacity at the same locations does not further reduce the risk
substantially, because the damage is primarily generated at other locations within the catchment. For changes in
land use (EL) and in vulnerability (V), median values of EAD increase from scenarios 0 to 2 (from €0.5 million

to €0.9 million). Similar increases are obtained for the component asset values (EA). These results imply that the
assumed changes in land use, asset values and vulnerability have considerable impacts on flood risk, only topped
by the change in dike heights.





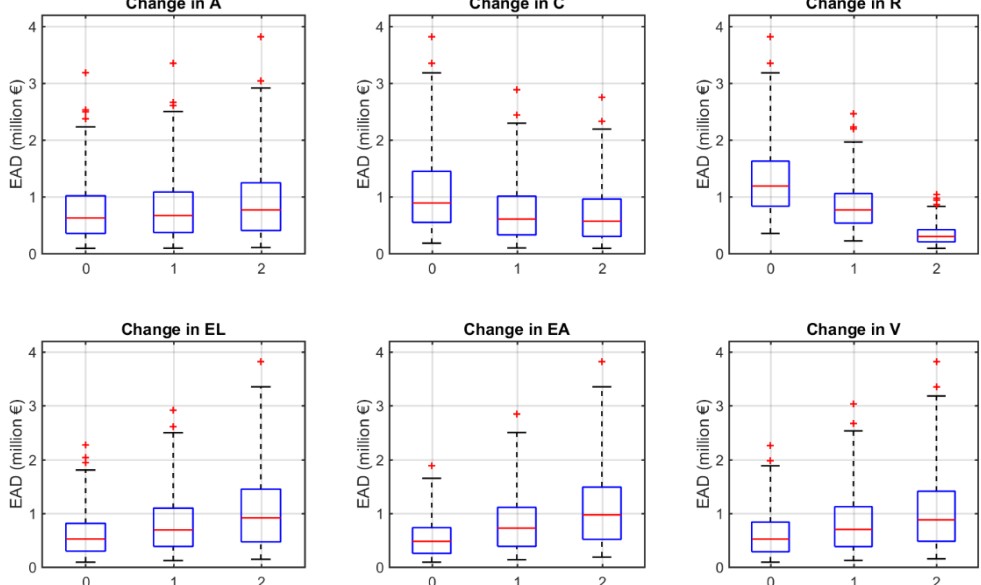

**Figure 5. Box plots of EAD, aggregated at the catchment scale, for changes in six components: atmosphere (A),**
**catchment (C), river system (R), land use (EL), asset values (EA) and vulnerability (V). The box plots show the median**
**values (red lines), the 25th and 75th percentiles (top and bottom of boxes) and the range (whiskers). Outliers are shown**
**by "+".**

Figure 6 shows the effects of the different components on the risk curve. This representation illustrates the effect
of changes in risk components across the whole spectrum of probabilities, whereas the EAD gives an aggregated
information. For each component, the baseline scenario is compared to the two symmetric scenarios, whereas only
the respective component is changed and all other components are fixed at their baseline state. The upper left plot
of Figure 6 shows the effect of change in the atmosphere (A). Differences between the risk curves are only visible
for high probability events, whereas for extreme events the risk curves are similar for different climate scenarios.
This is explained by the interplay of the flood regime in the Mulde catchment and the seasonal variations applied
in the climate change scenarios. Most of the floods occur in winter, however, the most extreme events tend to
occur in summer. Since the change scenarios, based on past observations, assume a strong increase in precipitation
in winter and almost no change in summer (see Table 2), climate change manifests itself mainly for high probability
events.

Changes in catchment (C) have the opposite effect on the risk curves, i.e. they affect only low probability events.
This is a consequence of the threshold process applied in the reservoir implementation in which the 100-year
discharge (HQ100) is used to cut off the extreme flood flow. Although reservoirs operated in this way are very
effective in reducing the peaks of extreme flood events, the reduction in EAD is modest compared to the effect of
other components, such as dike heightening. This can be explained by the small contribution of extreme events to
EAD. Merz et al. (2009) have shown that EAD is dominated by "high probability/low damage" events and that
"low probability/high damage" events play a small role, because their low probabilities overcompensate their high
damages. They have further argued that extreme events are more important for the affected societies than it is





expressed by their contribution to EAD. Hence, EAD is rather insensitive to changes in reservoir capacity in our case study, and the use of EAD as risk indicator might undervalue the risk reducing effect of reservoirs. This discussion also provides a note of caution on a higher level: the relative contribution of different components to

changes in risk varies across the probability spectrum, and changes that affect mainly low probability events may be undervalued by EAD which has been used almost exclusively in the studies to date (Table 1).

Changes in the river system (R) and in land use (EL) have substantial impact across the whole probability spectrum, whereas the impact of changes in asset values (EA) and in vulnerability (V) tend to increase from high probability to low probability events.

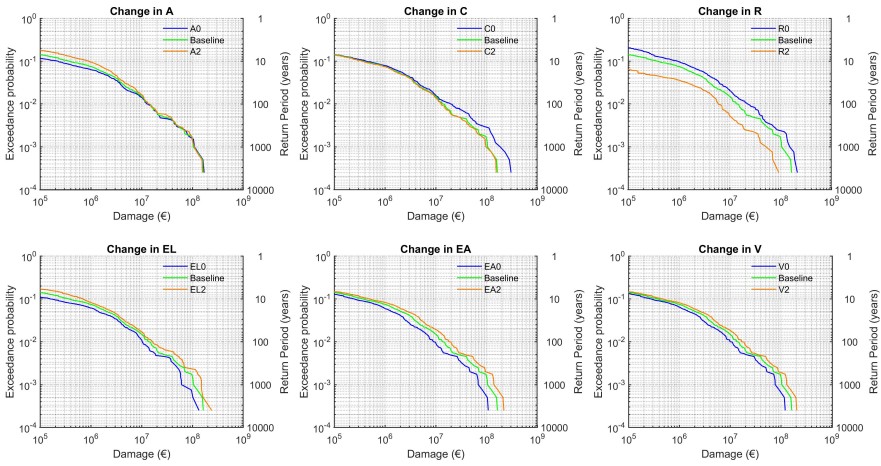


**Figure 6. Risk curves, for damages aggregated to the catchment scale, for changes in six components: atmosphere (A), catchment (C), river system (R), land use (EL), asset values (EA) and vulnerability (V) under baseline conditions. Baseline represents baseline scenarios for each component which is denoted by A1C1R1EL1EA1V1. All change scenarios vary only in the respective component. For example, A0 means A0C1R1EL1EA1V1.**

**4.2. Sensitivity of flood risk for selected upstream and downstream locations**

To get a better understanding of changes in risk and of their spatial heterogeneity within the catchment, two districts located upstream (Zwickau) and downstream (Anhalt-Bitterfeld) in the catchment are analysed in more detail. Their risk curves for changes in the six components, compared to the baseline, are given in Figure 7. The change in the atmospheric component (A) shows a similar behaviour in these two sub-basins as the whole

catchment. Regarding the change in catchment hydrology (C), change in flood storage capacity has a more dominant impact upstream which is explained by the reservoir locations. Due to the assumed reservoir operation the reservoir impact is only visible for very low probability events at the downstream sub-basin. Change in river system (R) strongly impacts risk both upstream and downstream. While the difference between scenarios with low dike height (R0) and baseline dike height (R1) is small upstream, there is a significant difference in the risk curves

between these scenarios at the downstream location for high probability events. One potential reason for this is the influence of topography on the number of exposed asset values. It is likely that under the assumption of equal value per exposed asset unit, steep upstream and flat downstream reaches are affected differently by the same flood





magnitudes. In flat downstream areas changes in dike heights result in great differences of damage values since more assets are flooded. From the risk curves of different land use scenarios, it should be noted that the increased
urban area scenario (EL2) increases risk upstream for high probability events and downstream for low probability events. The difference between EL0 scenario and EL2 scenario is high upstream for high probabilty events because reservoirs do not affect flows below the 100-year discharge. When they start to operate, risk for different land use scenarios becomes similar. However, the baseline land use scenario (EL1) and the EL2 scenario behave almost identical upstream which can be explained by the steep topography. The additional residential buildings for the
EL2 scenario might be located at steeper areas, and thus, they are not exposed to floods. On the other hand, the difference between the risk curves of EL1 and EL2 is high for extreme events at the downstream location. Risk curves of EL0 and EL1 scenarios are almost identical downstream. This can be explained by the specific setup of the residential buildings added in EL1 which are not exposed to floods. The last two components, change in asset values (EA) and vulnerability (V), have similar impact on the risk curves at both upstream and downstream
locations.

For the downstream district, abrupt (vertical) changes in the risk curves are observed around 500-year or greater return period events. In fact, events around this abrupt change have different peaks corresponding to different return periods but they show similar flood volumes. Therefore, they result in similar inundation depths and similar damage values for different probabilities.

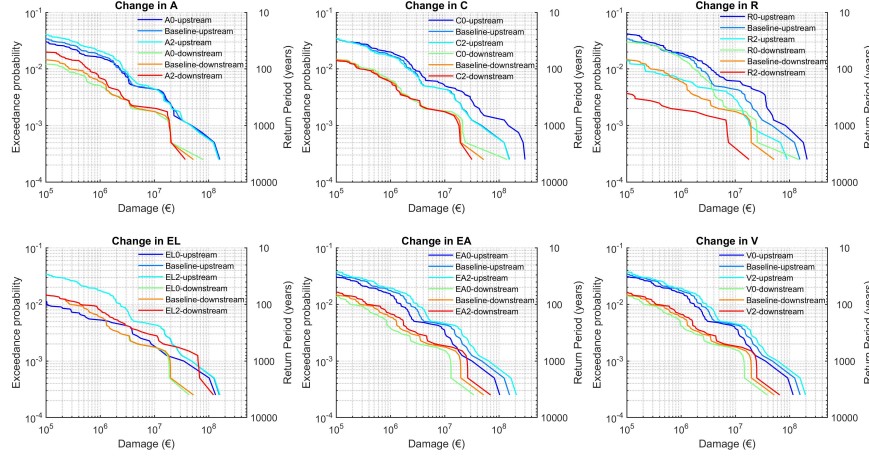


**Figure 7.** Risk curves for changes in six components: atmosphere (A), catchment (C), river system (R), land use (EL), asset values (EA) and vulnerability (V) under baseline conditions at districts Zwickau (upstream) and Anhalt-Bitterfeld (downstream).

**4.3 Seasonal effects on changes in risk curves**

To understand the temporal pattern of changes in risk, risk curves for summer and winter seasons are illustrated in Figure 8. Only the results for the atmosphere, catchment and river system components are shown, because they directly affect the peak flows in different seasons. It can be concluded that events in the summer season cause higher losses for the same return periods. We can observe different sensitivities in the winter and summer seasons.




First, for change in atmosphere (A), differences between change scenarios are observed throughout the whole probability range in the winter season. In summer, changes are very small. This is related to the much larger variation of precipitation values in winter compared to summer (Table 2). Second, change in catchment system (C) affects the risk curve for events with return periods higher than 500 years in winter, while differences can be observed already for the 100-year event in summer. This can be explained by the reservoir operation rule and the magnitude of events in different seasons. For example, the 100-year event in summer and the 800-year event in

winter are of similar magnitude corresponding to the 100-year flood of the annual time series, which is the threshold for reservoir operation. Finally, differences in risk curves across the whole probabilities range are visible for change in river system (R) for both seasons.

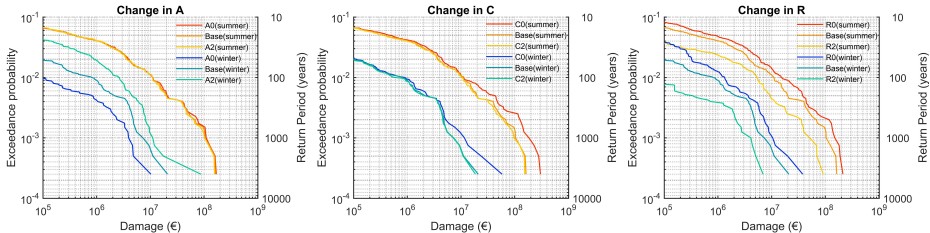

**Figure 8. Risk curves for changes in three components, atmosphere (A), catchment system (C) and river system (R),**
**under the baseline conditions for winter (blue colours) and summer (red colours).**

### 4.3. Relative influences of different components on flood risk

For a better visualization of the combined or opposed effects of different risk components on EAD, parallel-coordinates plots are used in Figure 9-11. These plots consist of seven parallel axes whereas the first six axes represent the different risk components, i.e. from left to right, changes in atmosphere (A), catchment system (C),

river system (R), land use (EL), assets (EA), and vulnerability (V). The seventh axis shows EAD obtained from different combinations of risk components: The scenarios are indicated by 0, 1 and 2 on the parallel coordinates, and each combination of components is colored according to its EAD value. In this way, combinations of risk components that result in a certain EAD interval are easily visualized.

In Figure 9 a subset of change scenarios is highlighted that result in very high EAD values above €2.5 million. It

is interesting to note that all these scenarios contain the low-dike height scenario (R0). As soon as another river system scenario (R1 and R2) is selected, EAD falls below €2.5 million. Increasing the dike height seems to be the most effective measure to keep the damage below a predefined threshold irrespective of changes in other risk components.





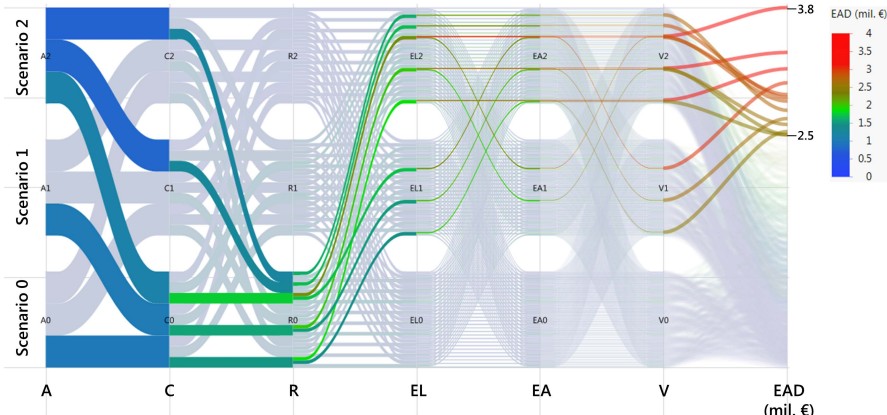


**Figure 9. Parallel-coordinates plot showing combinations of flood risk components that result in a certain EAD interval. From left to right, the six parallel coordinates represent changes in the flood risk components (A, C, R, EL, EA and V), and parallel coordinate on the right hand side shows EAD (mil. €) obtained from different combinations of risk components scenarios. Change scenarios are indicated by 0, 1 and 2 on the parallel coordinates.**


In order to understand the impact of climate change on EAD, the baseline scenario for all components and six different combinations with warmer climate scenario (A2) are analyzed (Figure 10). Particularly, we looked which other components can offset the effect of the atmospheric component. Under the fixed A2 scenario, five scenario combinations are highlighted, each time altering a different component from its baseline value towards EAD

decrease. For instance, in order to understand the relation between atmosphere and catchment changes, we compared the baseline scenario and the scenario of a warmer climate and increased storage capacity ($A2C2_1$), where subscript 1 denotes that all other components are kept in their baseline state. Scenario $A2C2_1$ causes an increase in EAD compared to the baseline EAD value meaning that climate change has a more dominant impact than catchment changes. Consequently, one could argue that changes in catchment system cannot compensate the

impact of climate change under the selected assumptions. In case of river system changes, $A2R2_1$ scenario decreases EAD to the value of €0.3 million, compared to the baseline scenario of €0.7 million. Hence, increased dikes can offset the adverse effect of the warming climate on flood risk. Changes in land use, asset values and vulnerability ($A2EL0_1$ $A2EA0_1$, $A2V0_1$) result in EAD below the baseline scenario thus compensating the effect of climatic changes.

To compensate the adverse effects of climatic changes, management options in all other risk chain components can be adopted. They are, however, associated with different implementation costs, different degree of feasibility or public acceptance. For instance, increase of dike heights along extended river networks can be very costly. Construction of additional reservoirs might adversely affect the ecological state of the river or be simply not feasible. We thus explored the set of scenarios, where changes in the catchment and river system were kept

constant. Asset values were kept at the baseline level or were allowed to increase. By changing the land use and vulnerability values, the EAD was retained in the range from €0.5 million to €2 million (Figure 11). Under these assumptions, it is possible to restrain the effect of climate change and increasing asset values on flood risk without implementing technical flood protection measures.

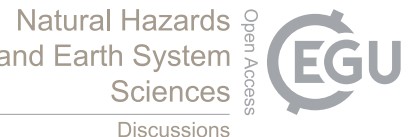
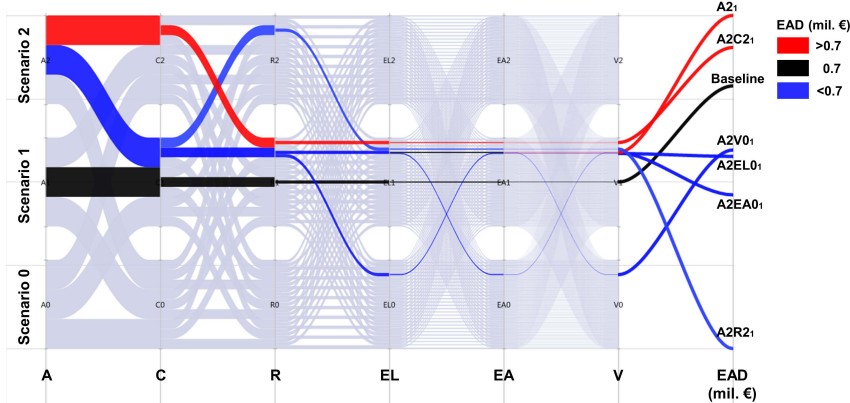

**Figure 10. Parallel-coordinates plot representing the baseline scenario (Scenario 1) for all components and six combinations of flood risk components with warmer climate scenario (A2): A2₁, A2C2₁, A2R2₁, A2EL0₁, A2EA0₁, and A2V0₁ where subscript '1' shows that all other unwritten components are in their baseline condition.**

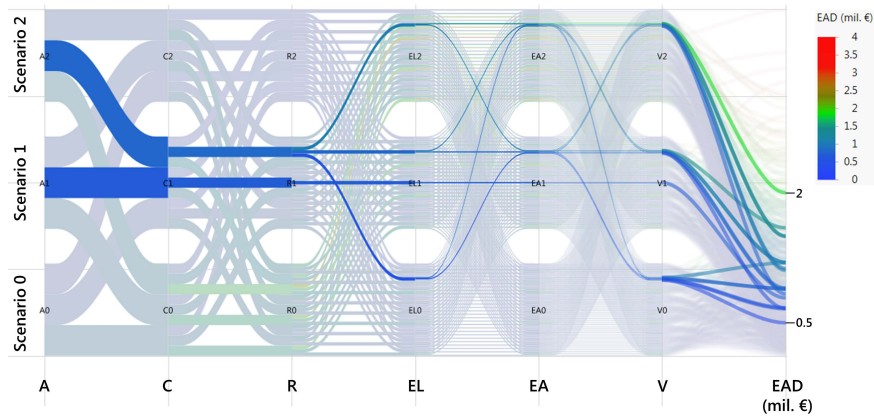

**Figure 11. Parallel-coordinates plot representing EAD for change in land use (EL) and vulnerability (V) under fixed baseline catchment and river system scenarios and increasing atmosphere and asset values.**

## 5. Discussion

The main purpose of this study is to fill the research gap on changes in flood risk, where consideration of the entire risk chain is generally missing. Taking into account all risk components allowed to explore the effect of changes in the individual risk chain components and their mutual interactions.

To the authors' knowledge, this study is the most comprehensive analysis on the influences of different drivers of flood risk including hazard, exposure and vulnerability drivers. The combination of sensitivity analysis with the DFRA approach overcomes a number of limitations of event-based risk assessments. Although our change scenarios have subjective assumptions, we used the best available data and options to create these scenarios. The



expected annual damage reaches a maximum of €4 million in our case, and for extreme events we obtain maximum
       absolute losses around of €100 million. For extreme events, changes in all risk components, except in the
       atmospheric component, have an impact on the damage. The impact of climate change is mostly visible for high
       probability flood events. This was explained by seasonal variations in precipitation change between scenarios in
       combination with the specific flood regime of the Mulde catchment.

The presented results are subject to limitations related to the flood risk chain model and the subjective assumptions
       for the reasonable change scenarios. Each model along the risk chain has limitations and uncertainties. For
       instance, water level calculation in the 1D hydrodynamic model strongly depends on river geometry estimated by
       the simplified river cross-sections. Neglected dike breaches (only overflow is considered) is another limitation in
       the representation of hydraulic processes. Further, flood damage estimation is sensitive to inundated areas and
exposed assets, both based on coarse DEMs. High uncertainties also pertain to flood damage modelling; they can
       have a larger contribution to uncertainties in risk estimates than uncertainties in hydrological/hydraulic
       components (Apel et al., 2009; de Moel and Aerts, 2011; Vorogushyn et al., 2012). More detailed discussion on
       limitations of the flood risk model chain can be found in Falter et al., (2016).

       The impact on flood risk highly depends on the defined change scenarios of the risk components. In the sensitivity
analysis, there is some subjectiveness in their selection. The assumed change amounts for each component and the
       methods to create plausible change scenarios reflect different subjective choices. For instance, the climate change
       scenarios were generated based on observed past changes. Due to anthropogenic climate change, the effects on
       temperature and precipitation will likely be different. However, in order to explore the effect of reasonable changes
       in climate on flood risk, we consider this assumption acceptable, as this study does not attempt to evaluate flood
risk under various climate projections available to date. In the catchment change scenarios, we used large changes
       such as doubling the reservoir storage capacity. Yet, we observed comparatively small effects for the particular
       case study area given the implemented operation rules. Scenarios for river system were determined based on
       possible changes in dike heights adopted from the literature. Conditional on our assumptions, change in dike height
       is able to compensate the risk-increasing impact of other components. Land use change scenarios were created
based on increase in residential areas between the years 1990 and 2012 by randomly changing the state of single
       pixels. The selection of the time period as well as the spatial distribution of changes in individual pixels is
       obviously subjective. The latter can potentially be overcome by considering multiple scenarios of spatial
       distribution of changes in pixel state in relation to distance to the river and thus propensity for inundation. In the
       vulnerability scenarios, we only focused on the impact of private precautionary measures. Other aspects, such as
awareness and preparedness, can also alter vulnerability. However, between the disastrous floods in 2002 and 2013
       in Germany, private households and companies substantially adopted precautionary measures (Kreibich et al.,
       2017). Therefore, our scenarios are reasonable to represented changes in vulnerability.

       These subjective assumptions do not influence the main conclusion of our study, namely the need to analyse
       changes in flood risk by considering the whole range of drivers. This effort is still to be undertaken to fully
understand the risk and to devise appropriate measures for risk reduction going beyond technical flood protection
       and focussing only on adverse consequences of climatic changes. Using the proposed blue print, the effect of
       different measures under more elaborated and specific assumptions can be explored at other sites, possibly
       accompanied by cost-benefit analyses.



**6. Conclusions**

In this study, a comprehensive sensitivity analysis was performed considering six different components related to hazard, exposure and vulnerability. The sensitivity analysis was combined with the 'Derived Flood Risk Analysis based on continuous simulation (DFRA)' proposed by Falter et al. (2015). This framework was applied to the mesoscale Mulde catchment in Germany in order to explore the effects of plausible changes in flood risk chain components on risk estimates and to understand interactions between different components.

Our study finds that the largest contribution to flood risk changes comes from the change in river system considering heightening of river dikes. In this case, EAD (Expected Annual Damage), aggregated at the catchment scale, is at most €1.1 million. Interestingly, climate change impacts would be offset by these river system changes. However, dike rising might not be a feasible option because it is costly, requires space, and has long implementation times. Alternatively, changes in land use and vulnerability could be considered to reduce economic damage and were shown to be capable to compensate adverse impacts of climatic changes. In terms of feasibility, vulnerability reduction is more realistic; decrease in settlement areas is a long-term approach and rarely implemented even in high flood-prone areas, as additional factors besides the actual flood risk play a role in the decision to resettle an area. The effect of climatic changes on flood risk is modest in our setting. This is a consequence of climatic changes being out of phase with flood generation: Large floods occur in summer where precipitation change is small. The majority of floods occur in winter where climatic change is substantial, however, these floods are typically small and do not cause large damage. Change in catchment system has a visible impact in the upstream reaches because most of the reservoirs are located there. Implementing storage capacity has a surprisingly modest effect on EAD. This results from the operational setting, as only floods higher than the 100-year event are influenced by the reservoirs, and the fact that EAD is typically dominated by the contribution of smaller floods.

Although the results are specific to the case study and depend to some extent on our choices in the implementation of this framework, some general conclusions can be derived:

1. The risk, quantified as EAD (Expected Annual Damage), varied by a factor of 40, from €0.1 million to €4 million, across the range of change scenarios. This is a very high variation given the fact that our change scenarios represent possible changes that can occur within a few decades. This result points to the significant volatility that can be associated to flood risk. It underscores the necessity to monitor changes in risk regularly.

2. Our literature analysis revealed that past studies on changes in flood risk have almost exclusively focused on effects of climate change and land use change. Our analysis demonstrates that other components that have been neglected can be even more important. Hence, the study calls for more comprehensive analyses of changes in flood risk.

3. The effects of external drivers, i.e. drivers which cannot be controlled within the catchment (in our case climate change and increase in asset values) can be offset by internal factors. This points to the options of local stakeholders to counteract flood risk growth due to climate change and economic growth by flood risk management.

4. Almost all past studies on changes in flood risk have used EAD as risk indicator. Since EAD is typically dominated by the contribution of small and medium floods, management options which reduce the damage



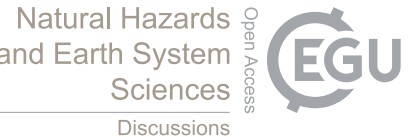

for large floods are penalised by this limitation to EAD. A more comprehensive investigation, e.g. by considering effects across the risk curve, seems necessary.

**Acknowledgements.** We would like to acknowledge funding from the European Union's Horizon 2020 research and innovation programme under the Marie Skłodowska-Curie grant agreement No 676027.

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





**Tables**

**Table 1: Simulation-based studies on the causes of flood risk changes and their relative contributions. H, E indicate whether changes in hazard or exposure are investigated. (EAD: Expected Annual Damage; EAP: Expected Annual Population exposed).**

| Study | Time frame, region | Climate change (H) | Land subsidence (H) | Change in GDP (E) | Change in population (E) | Change in asset values (E) | Change in land use (E) | Change in cropland area (E) | Risk indicators | Dominant drivers of change in flood risk |
|---|---|---|---|---|---|---|---|---|---|---|
| | | **Drivers considered** | | | | | | | | |
| **Alfieri et al. (2015)** | 1990-2080, Europe (28 countries) | ✓ | | ✓ | ✓ | | | | EAD, EAP | ▪ Combinations of change in climate, in GDP and in population |
| **Arnell and Gosling (2016)** | 2050, global (20 regions) | ✓ | | ✓ | ✓ | ✓ | | ✓ | EAD, EAP | ▪ Climate change |
| **Bouwer et al. (2010)** | 2040, south Netherlands | ✓ | | | | ✓ | ✓ | | EAD, Loss probability curves | ▪ Climate change |
| **Budiyono et al. (2016)** | 2030, Jakarta | ✓ | ✓ | | | | ✓ | | EAD | ▪ Land subsidence and land use change |
| **Elmer et al. (2012)** | 1990-2020, Mulde River, Germany | ✓ | | | | ✓ | ✓ | | EAD | ▪ Land use change |
| **Feyen et al. (2009)** | 2071-2100, Europe | ✓ | | | | | ✓ | | EAD | ▪ Land use change |



| | | | | | | | | |
|---|---|---|---|---|---|---|---|---|
| **Feyen et al. (2012)** | 2071-2100, Europe | ✓ | | | | | | EAD, EAP | ▪ Climate change |
| **Hall et al. (2003)** | 2030-2100, England and Wales | ✓ | | ✓ | ✓ | ✓ | ✓ | EAD, EAP | ▪ Change in GDP, asset values, land use and population (socio-economic drivers) |
| **Hatterma nn et al. (2014)** | 2011-2100, Germany | ✓ | | | | | | EAD | ▪ Climate change |
| **Lung et al. (2013)** | 2011-2040 and 2041-2070, Europe | ✓ | | | | ✓ | ✓ | 3 indicators related to 100-year flood: percentage of flooded area; mean water depth of flooded area; percentage of commercial & industrial areas within flooded area (only for 2011-2040) | ▪ Combinations of change in climate, in asset value and in land use |
| **Muis et al. (2015)** | 2000-2030, Indonesia | ✓ | | | | | ✓ | EAD | ▪ Land use change |
| **Rojas et al. (2013)** | 2000-2080, European Union | ✓ | | ✓ | ✓ | ✓ | | EAD, EAP | ▪ Change in GDP, asset values and population (socio-economic drivers) |
| **Te Linde et al. (2011)** | 2030, Rhine catchment | ✓ | | | | | ✓ | EAD | ▪ Climate change |





**Table 2: Baseline and change scenarios for the sensitivity analysis. For each component the variables that are changed in the sensitivity analysis and their scenario values (S1: baseline; S0, S2: change scenarios) are given.**

| Component | Variable | Scenario values(S0 / S1 / S2) | Explanation |
|---|---|---|---|
| **Atmosphere (A)** | Precipitation [mm] | Winter: (-19.0 / 0 /+19.0)<br>Spring: (-8.1 /0/+8.1)<br>Summer: (+1.1/ 0 / -1.1)<br>Autumn: (-5.9 / 0 /+5.9) | Daily precipitation is multiplied by change factor $(1 + \Delta_p/\overline{p^0})$ where $\overline{p^0}$ is the mean precipitation amount for the baseline scenario series and $\Delta_p$ is the seasonal change in mean precipitation over the 50 years period. $\Delta_p$ values are given in the third column. |
|  | Temperature [°C] | Winter: (-0.49 / 0 / +0.49)<br>Spring: (-0.45 / 0 / +0.45)<br>Summer: (-0.45 / 0 / +0.45)<br>Autumn: (-0.38 / 0 / +0.38) | Change in mean temperature over the 50 years is added to daily temperature value on seasonal basis. |
| **Catchment (C)** | Reservoir capacity [Mio m³] | 0 / 106 / 212 | Current capacity is doubled and completely removed. |
| **River system (R)** | Dike height [m] | (-0.5 m / 0 / +0.5 m) | Current dike height is changed by 0.5 m. |
| **Land use (EL)** | Residential area [km²] | 560 / 672 / 784 | Current residential land use area is changed by 112 km². |
| **Value of assets (EA)** | Building price index | 0.66 / 1 / 1.34 | Current index is changed by 34 %. |
| **Vulnerability (V)** | Scaling factor of relative damage | 0.71 / 0.95 / 1.20 | Scaling factor of medium level precaution is increased and decreased by 26 %, for the cases of no precautionary measure and high precaution level, respectively. |