# Peer review of "How do changes along the risk chain affect flood risk?"

_Natural Hazards and Earth System Sciences, 2018_

## Referee Comment (RC1) · Anonymous Referee #1 · 5 Jul 2018

The paper studies the sensitivity of flood risk to various factors: Changes in precipitation and temperature, reservoir size, dike height, distribution of residential areas, the value of affected buildings and private household precautions. Thus, the factors consider changes in climate, catchment, river system, land use, assets and vulnerability. Changes in the likelihood of monetary losses are determined by changing the aforementioned factors. To achieve this, a model chain consisting of a weather generator, a rainfall-runoff model, a river-network routing model, a hinterland inundation model as well as a flood-loss model is set up. For each of the 729 selected scenario combinations 4000 years of continuous model simulations are produced and analysed.

The study is within the scope of NHESS. It is important to the scientific community as it shows that it is possible to consider the entire flood risk chain and identify the

most influential factors for flood risk. It is the first/one of the first studies attempting this comprehensive approach. The most important result is that, not-surprisingly, increases in flood risk due to climate change can be compensated by appropriate protection and precautionary measures.

Major comment:

The changes imposed on the different risk components are very different in magnitude (e.g. small atmospheric changes but huge changes in reservoir size (+100%)). To get a better idea of the sensitivity of losses to these components it would be good to add a graph which displays the influence of normalised changes on the losses. I am aware of the fact that there will be some subjectivity when deciding on a suitable way to normalise the imposed changes.

Minor comments:

p.1 l.38 Is Kreibich et al. the first/only publication that states this? Otherwise please add. "e.g." to the reference. Please also check this for all other references.

p.2 l.78 Did you search for all keywords separately or only in combination?

p.9 l.231 "More importantly, with the new setup, the SWIM model seems to be able to represent the cut-off process more accurately." Where do I see this?

p.10 l.258-266 For which size of area is the model acceptable (what is meant by "large-scale")?

p.11 l.271-272 Please specify: Less damage if people are regularly affected.

p.11 l.280-281 How do modelled return periods compare to observed return periods?

p.11 l. 283-287 Are these values better when reservoirs are considered? (You show before that discharge was improved.)

p. 11 l.298 It would be interesting to learn something about the computational demand

of the different modules of the model chain, relative to each other (e.g. 10% weather generator, 30% SWIM ....)

p.12 Fig. 4 In my opinion this figure is not needed.

p.12 l.336-339 Please give the numbers or refer to table 2.

p.13 sec. 3.23 Please move the explanation on the operation of reservoirs from page 15 to this section.

p.14 l.402-403 In the discussion section you clarify that anthropogenic climate change may be associated with very different temperature and precipitation values (patterns, duration, clustering ....) in the future, which are not captured by your approach. Here you seem to suggest that the other studies overrate the effect of climate change. Please rephrase.

Figs 9+11 I find it impossible to follow the path of the thin lines towards the right of the plot. I assume that this could be improved if you consider to refrain from changing the colours along the graph. In case you want to keep the colour change you need to explain it in the text. It took me a long time to figure out what the colour change (probably) wants to tell me. I would also suggest to add a black default-scenario line.

Technical remarks:

p.2. l.69 Something is missing in this sentence

p.4 l.137 "but they also have storage capacity"

p.4 l.144 "the Mulde catchment has been hit by large floods associated with high damages before"

Fig. 10: EAD values on y-axis are missing

p.21 l.592 "our scenarios are reasonable to represent"
* * *
2018-155, 2018.

---

## Referee Comment (RC2) · Anonymous Referee #2 · 13 Jul 2018

Journal: NHESS
Title: **How do changes along the risk chain affect flood risk?**
Author(s): Metin et al.
MS No.: NHESS-2018-155
MS Type: Research Article
**Iteration: First review**

The objective of the paper is to contribute in filling the gap in current understanding of the roles of the different components of the risk chain on changes in flood risk. To this aim a simulation-based approach is implemented where, starting from the investigation of a baseline scenario, six components of the risk chain are changed (namely: climate change, implementation of reservoirs in the catchment, flood protection along the rivers, land use change, change in asset values and changes in the vulnerability of flood-affected objects) by both increasing and decreasing their reference value. Thanks to the implementation of the DFRA, this allows to simulate 729 damage series (of 4000 years) from which the EAD and the risk curve are derived and investigated: (i) at the catchment scale, (ii) at two typical upstream and downstream sub-basins, and (iii) for summer and winter seasons.

The topic of the paper is in the scope of NHESS and results can contribute to a better understanding of flood risk and risk mitigation strategies, also in the light of future climate change. The paper is overall clear, well-structured and well-written; conclusions are mostly supported by simulation results. Still, there are some points that need further clarification or that must be better explained in order to make the paper totally understandable; these are reported as "specific criticisms". On the other hand, in the following, some suggestions are reported that could increase the robustness and the completeness of the research.

**Suggestions**

- The choice of considering a double storage capacity as the change in the catchment hydrology is not totally clear to me. In the light of choosing "plausible deviations from the baseline", it makes more sense considering a change in the operational rules like, for example, in the value of the cut-off discharge. I suggest authors to explore also this scenario;
- Again, in the light of choosing "plausible deviations from the baseline", also the change in building quality should be investigated. This is a quite cheap strategy for risk mitigation that can be easily encouraged/achieved by public and private incentives. I suggest authors to explore also this scenario;
- The whole analysis is based on the estimation of damage to the only residential sector. Still, risk can be heavily affected by damage to other sectors like agriculture, commerce, tourism, population, etc. For some of these sectors flood damage is strongly related to the season of occurrence of the event (e.g. agriculture, tourism), and it may be the case that the effect of climate change on such sectors modifies present conclusions on EAD and on the role of the different components of the risk chain. I think that more than one sector should be included in the analysis or, at least, some considerations must be added on the possible role of damage to other exposed sectors.

**Specific minor comments (which can increase the readability and clarity of the paper)**

Section 1

Pg. 2 line 65 "A major problem is the superposition of several drivers of risk changes" → what authors mean here with "superposition"? Please, specify

Section 3

Pg. 6 line 164 "the approach provides the complete flood hydrograph" → on a daily base, is it correct?

Pg. 6 line 166 "the spatial dependence between flood damages at different locations in the catchment is taken into account" → what authors mean with "spatial dependence between flood damages at different locations"? Please, specify

Pg. 7 line 183 "The weather generator is parameterized on a monthly basis." → this is already stated in the following page. The sentence can be deleted

Pg. 8 lines 215 -221 → Some assumptions made for the modelling of reservoirs are not totally clear to me: (1) are reservoirs empty at the begging of the simulation? If yes, is it realistic? If no, which is the initial volume? Why? (2) What happens if the storage capacity is reached before the discharge falls below HQ100? (3) Why the return period of 100 years was chosen? Is it the design return period of dikes? (4) Which is the necessary information collected from Sächsisches Landesamt für Umwelt und Geologie?

Pg. 9 line 226 "The calibration and validation results illustrate obvious improvement in this new model setup compared to the version used in Falter et al. (2015)" → I cannot appreciate this improvement in Figure 3. On the left, only results from the new model and observations are reported so I cannot see the difference between the two models. Graphs on the right suggest that the two models are mostly equivalent. Please, comment on this

Pg. 9 line 231 "with the new setup, the SWIM model seems to be able to represent the cut-off process more accurately" → Of course, the old model did not consider cut-off

Pg. 10 line 254 "the minimum height was assumed at 1.8 m" → on which bases?

Pg. 10 line 255 "the 2D raster based model uses a 100 m resampled computational grid from DEM10, which was found an acceptable compromise for representation of inundation characteristics and computation time" → I think some considerations must be included on topography. 100 m can be enough in flat areas (i.e. downstream) but can introduce big errors in damage estimation in steep areas (i.e. upstream). Did authors consider different resampling of the DEM in different areas of the catchment?

Pg. 10 line 265 "Although there is an underestimation of inundation, the model gives a reasonable estimate of inundation extent and depth for large-scale assessments" → which are the bases for this statement? 50% underestimation in flood extent is a significant error in my point of view

Pg. 11 lines 275-282 → I think that assumptions made for the estimation of damage must be better explained: (1) which is the scale of analysis? The 100*100 m2 cell? The municipal scale? Other? (2) how building type and level of precaution are assessed? (3) do asset values depend on building quality and type?

Pg. 11 line 283 "The sum of damages for all communities was officially reported as €240 million" → does it refer to the total damage or damage to residential buildings?

Pg. 11 line 285 "This underestimation may be explained by uncertainty in asset values and their spatial distribution and uncertainty in the damage model" → and underestimation of flood extend I guess

Pg. 11 line 302 "For example, the baseline scenario of the catchment component is represented by a model version calibrated for a recent time period and including the current implementation of reservoirs in the catchment. The specific time periods and assumptions for the baseline scenarios are given in sections 3.1.1 to 3.1.4 where the implementation, calibration and validation of the different modules for the current situation are described" → the meaning of the baseline scenario is clear to the reader at this point of the paper. This sentence can be omitted.

Pg. 13 line 356 – 360 → Are studies made in the Netherland transferable to the Mulde catchment? What authors mean with "potential" dike heightening? Potential with respect to what?

Pg. 13 line 364 "The change scenario EL2 is based on the increase in area of these two classes from 672 to 784 km2 between 1990 and 2012 where the change area was added to baseline scenario" → How the urban area was changed? I can understand this only at pg. 21

Pg. 13 line 367 "Pixels (100 x 100 m2) of the classes 111 and 112 were assigned to non-residential land cover classes (i.e. agricultural areas and semi-natural areas)" → how pixels were re-assigned? Why authors did not consider CORINE land use map of 1990? I think it is more realistic.

Section 4

Pg. 14 line 405 "This non-symmetry in the effects of the catchment component is explained by the specific implementation of the reservoir capacity: Implementing a capacity of 106 million m3 reduces the EAD significantly, but doubling this reservoir capacity at the same locations does not further reduce the risk substantially, because the damage is primarily generated at other locations within the catchment" → not clear, the role of reservoirs is not reducing damage downstream? Please, clarify

Pg. 16 line 455 "Regarding the change in catchment hydrology (C), change in flood storage capacity has a more dominant impact upstream which is explained by the reservoir locations. Due to the assumed reservoir operation the reservoir impact is only visible for very low probability events at the downstream sub-basin" → I still do not understand the influence of reservoirs in the catchment. Readers should be supported by a better description/discussion of the location of reservoirs with respect to the sub-basins.

Pg.17 line 464 "From the risk curves of different land use scenarios, it should be noted that the increased urban area scenario (EL2) increases risk upstream for high probability events" → I cannot see the difference between EL2 and the baseline scenario in Figure 7. Is one curve missing?

Pg. 17 line 468 "the baseline land use scenario (EL1) and the EL2 scenario behave almost identical upstream which can be explained by the steep topography" → I guess it depends on the rules adopted for increasing the urban area and on how the flood extent changes for different return periods

Pg. 17 line 472 "This can be explained by the specific setup of the residential buildings added in EL1 which are not exposed to floods." → not clear, please specify

Pg. 19 line 523 "Under the fixed A2 scenario, five scenario combinations are highlighted, each time altering a different component from its baseline value towards EAD decrease" → I can see four combinations leading to lower EAD. Could authors check?

Figures

Figure 1 – subcatchments are not visible in mountain areas

Figure 2 – (1) please specify what authors mean with XS profile (2) output of the flood loss model is missing (3) level of precaution and contamination are missing in the box related to FLEMOps

Figure 4 – I think that the figure is not explicative of the logic tree. Please, consider changes.

Figures 6, 7 and 8 are too small

Bibliography

I did not check the bibliography at this stage of the review. I reserve to do this in a second time.

---

## Author Response (AR1)

[revised manuscript text omitted]

Legend

**Reservoirs**

**Max. Storage Cap. (mil. m3)**

- 0.01 - 1.5
- 1.51 - 5.0
- 5.1 - 10.0
- 10.1 - 25.0
- 25.1 - 55.0
▲ Gauges
Residential Area
River
Selected Districts
Subcatchment

**Elevation (m a.s.l)**

- 52-250
- 260-500
- 510-750
- 760-1200

Anhalt-Bitterfeld (downstream)

Zwickau (upstream)

BAD DUBEN

GOLZERN 1

ERLLN

WECHSELBURG 1

ZWICKAU-POELBITZ

Vereinigte Mulde

Freiberger Mulde

Zwickauer Mulde

Zschopau

Berlin

Dresden

Köln

Frankfurt

München

0  10  20  30  40  Kilometers

[Figure]

**Fig. 1. Study area Mulde catchment, including main tributaries, reservoirs and river gauges. The inset shows the location of the catchment within Germany.**

**3. Methods**

**3.1. Flood risk simulation model chain**

To simulate the complete flood risk chain, the Regional Flood Model (RFM) is used. RFM consists of a weather generator, rainfall-runoff model, 1D channel routing model, 2D hinterland inundation model and flood loss estimation model for residential buildings. The results of one model are used as input for the next model. Fig. 2 shows the model chain and gives the most important information on the input data and the characteristics of the different modules. Details about the model chain are given in Falter et al. (2015). The computational demand of the different modules is as follows: 8% RWG (coverage: Germany+), 10% SWIM, 80% RIM, 2% FLEMOps. Please note that RIM runs on a mixed infrastructure CPU + GPU. The other components run on CPU only.

The model setup follows the concept of derived flood risk analysis based on continuous simulation proposed by Falter et al. (2015). A weather generator provides spatially consistent meteorological fields which propagate through the entire model chain. In our study, the chain is run on a daily time step for 40 realizations of 100 years resulting in a total time series of 4000 years. Risk estimates are then derived directly from the time series of damage generated by the model chain.

A derived flood risk analysis based on continuous simulation has a number of advantages compared to event-based flood risk estimates. For instance, due to the continuous simulation the antecedent catchment conditions are implicitly considered in the flood generation, and the approach provides the complete flood hydrograph on a daily base. Since all models within the chain are spatially explicit, the approach provides spatially consistent flood events including the river-floodplain and damage processes. Hence, also spatial consistency of losses across the spatial dependence between flood damages at different locations in the catchment is taken into account. A further advantage is that risk is estimated using the space-time fields of damage. Hence, this approach follows the definition of risk, where risk is understood as the probability of exceeding a given damage. In contrast, traditional flood risk analyses use the probability of discharge as proxy for the probability of damage. For a comprehensive discussion see Falter et al. (2015).

Note that our model setup is the same as in Falter et al. (2015). The only difference is that we consider reservoirs in the rainfall-runoff module. The different modules along the risk model chain are described in the following.

**BASIC INPUT**

**COMPONENT**

**SPATIALITY**

Hydro-meteorological data for training the model: precipitation, temperature, etc

**Regional Weather Generator (RWG)**

multi-variate, multi-site, mixed distribution, auto-correlation, spatial correlation, daily resolution, regional scale

station-based

long-term spatial consistent meteorological series

Land use data, soil data, Digital elevation model (DEM)

**Rainfall-Runoff Model (SWIM)**

Subbasin-wise routing of daily discharge with Muskingum method

subbasin-based

long-term spatial consistent river discharge series

Channel's network, XS profile and roughness distribution

**Regional Inundation Model (RIM)**

**River network routing: 1D channel network model**

Routing flow along the river network in case of dike overtopping, computation of dike overflow with weir equation

river reaches

hinterland inundation level

dike overtopping discharge

DEM and roughness distribution

**Hinterland Inundation: 2D Raster-based Inertial model**

Simulation of hinterland inundation and calculation of inundation depth

raster-based

Inundation depth, duration

Asset values, land use data and building characteristics

**Flood Loss Model (FLEMOps)**

Calculation of direct damage to residential buildings on basic of water depth, return period, building type and quality

raster-based

[revised manuscript text omitted]

**Reply to Referee #1**

First of all, we would like to thank the referee for the time and effort put into reviewing the manuscript. In the following, we provide our responses to the individual questions posed by the referee.

**The paper studies the sensitivity of flood risk to various factors: Changes in precipitation and temperature, reservoir size, dike height, distribution of residential areas, the value of affected buildings and private household precautions. Thus, the factors consider changes in climate, catchment, river system, land use, assets and vulnerability. Changes in the likelihood of monetary losses are determined by changing the aforementioned factors. To achieve this, a model chain consisting of a weather generator, a rainfall-runoff model, a river-network routing model, a hinterland inundation model as well as a flood-loss model is set up. For each of the 729 selected scenario combinations 4000 years of continuous model simulations are produced and analysed.**
**The study is within the scope of NHESS. It is important to the scientific community as it shows that it is possible to consider the entire flood risk chain and identify the most influential factors for flood risk. It is the first/one of the first studies attempting this comprehensive approach. The most important result is that, not-surprisingly, increases in flood risk due to climate change can be compensated by appropriate protection and precautionary measures.**

We would like to thank the reviewer for his/her positive feedback. We are pleased that the reviewer finds the research important for scientific community.

**Major comment:**
**The changes imposed on the different risk components are very different in magnitude (e.g. small atmospheric changes but huge changes in reservoir size (+100%)). To get a better idea of the sensitivity of losses to these components it would be good to add a graph which displays the influence of normalized changes on the losses. I am aware of the fact that there will be some subjectivity when deciding on a suitable way to normalise the imposed changes.**

We thank the reviewer for this comment. Although the changes imposed on the different risk components are very different, we would like to highlight that the imposed changes are plausible. In response to this comment, we add a graph which shows percentage changes in the different components versus percentage changes in the median expected annual damage (EAD). It is visible that river system has a big impact on loss despite the fact that the imposed change is one of the lowest. However, the normalization to obtain percentage change would be very subjective and may mislead the readers. For example, adding 0,5 °C during the winter season would mean a large change of 100%, whereas adding the same delta during the summer would be a small change of 3%. Summing up these different changes for the years does not provide a sensible piece of information.  Further, the imposed changes depend on the spatial characteristics of the component which is changed in the sensitivity analysis. For instance, for atmosphere the change is imposed on all sub-basins across the catchments, whereas for the catchment component, it depends on the existence and the size of the reservoirs. Therefore, we prefer not to add this figure to the manuscript as it could easily mislead the reader and we do not see that it provides additional, useful information.

[Figure]

**Minor comments:**

**p.1 l.38 Is Kreibich et al. the first/only publication that states this? Otherwise please add. "e.g." to the reference. Please also check this for all other references.**

It is not the first/only publication. We added 'e.g.' to the reference and checked for all other references.

**p.2 l.78 Did you search for all keywords separately or only in combination?**

We searched all keywords both in combination and separately, not to miss any relevant study.

**p.9 l.231 "More importantly, with the new setup, the SWIM model seems to be able to represent the cut-off process more accurately." Where do I see this?**

We thank the reviewer for this comment (which has been addressed by the reviewer #2 as well). Because the reservoir component is only present in the new setup of the SWIM model, we removed the irrelevant text "more accurately" from the sentence. Additionally, we will also modify the paragraph to make it clearer.

**p.10 l.258-266 For which size of area is the model acceptable (what is meant by "largescale")?**

The model has been developed for the risk assessment in large-scale catchments, but the quality of model is adequate for the Mulde catchment, which is a meso-scale catchment.

**p.11 l.271-272 Please specify: Less damage if people are regularly affected.**

We included this statement to support that's why advanced version of FLEMO considers the return period of the inundation.

**p.11 l.280-281 How do modelled return periods compare to observed return periods?**

We did not compare modelled return periods with observed return periods directly. However as stated in the manuscript, the SWIM model has been validated against daily stream flows with a focus on high flow at selected gage stations. This implies that the model is supposed to be adequate for modelling flood (extreme) events and hence the return periods should be reasonably modelled by the hydrological model. The FLEMO damage model uses three classes of flood return period as descriptive variable (<10 year, 10-99 years, >100 years) (Elmer et al., 2010). Therefore, we believe that the final risk estimates are not too sensitive to small deviations in the estimated

return periods. Moreover, in the sensitivity analysis presented here, relative changes in risk are in focus rather than accurate estimates of the actual risk.

**p.11 l. 283-287 Are these values better when reservoirs are considered? (You show before that discharge was improved.)**

Actually, the mismatch is larger when the reservoirs are added into the model. The calculated damage amounts to approximately €61 million and is slightly reduced due to retention effect. The major problem is the mismatch between observed and simulated inundation areas due to dike breaches. Dike failures are insufficiently implemented and captured in the model (only dike overflow is considered). We will mention the change in loss estimation due to reservoir implementation.

**p. 11 l.298 It would be interesting to learn something about the computational demand of the different modules of the model chain, relative to each other (e.g. 10% weather generator, 30% SWIM ....)**

Approximately computational demand: 8% RWG (coverage: Germany+), 10% SWIM, 80% RIM, 2% FLEMOps. Please note that RIM runs on a mixed infrastructure CPU + GPU. The other components run on CPU only.

**p.12 Fig. 4 In my opinion this figure is not needed.**

Figure 4 shows the combinations of three plausible change scenarios for each of the six components. We added this figure to show the complexity in the design of the scenarios and to help readers to capture the idea. By considering Referee #2's comment on this figure as well, we propose to change caption of the figure. We shall use "conceptual scheme" instead of "logic tree".

**p.12 l.336-339 Please give the numbers or refer to table 2.**

We will refer to Table 2 in the revised version.

**p.13 sec. 3.23 Please move the explanation on the operation of reservoirs from page 15 to this section.**

We thank the reviewer for this comment. In the page 8, l.215-220, we have explained the operation of reservoirs. Therefore, we would prefer not to repeat it here. Instead, we will move the explanation on the operation of reservoirs in the page 15, 'Reservoirs operated in this way are very effective in reducing the peaks of extreme flood events' to p.8 l.220.

**p.14 l.402-403 In the discussion section you clarify that anthropogenic climate change may be associated with very different temperature and precipitation values (patterns, duration, clustering ....) in the future, which are not captured by your approach. Here you seem to suggest that the other studies overrate the effect of climate change. Please rephrase.**

We thank the reviewer and we rephrased this sentence as follows. "Another remarkable result is the rather small increase in the median loss values for changes in the atmosphere (A) from scenarios 0 to 2 (from €0.6 million to €0.8 million), despite the realistic assumptions on average changes in climate variables. Although our model does not capture complex change patterns such as changes in duration of wet spells or clustering of events, this result indicates that changes in climate might be not the dominant ones along the risk chain contrary to the prevailing perception."

**Figs 9+11 I find it impossible to follow the path of the thin lines towards the right of the plot. I assume that this could be improved if you consider to refrain from changing the colours along the graph. In case you want to keep the colour change you need to explain it in the text. It took me a long time to figure out what the colour change (probably) wants to tell me. I would also suggest to add a black default-scenario line.**

We improved these figures by using same colour along the graphs.

**Technical remarks:**
**p.2. l.69 Something is missing in this sentence**

We rearranged this sentence as follows. "Hence, conclusions from normalization studies, such as there is no evidence for the effect of human-induced climate change on the loss trend (e.g. Barredo, 2009), need to be taken with care."

**p.4 l.137 "but they also have storage capacity"**

Corrected.

**p.4 l.144 "the Mulde catchment has been hit by large floods associated with high damages before"**

Corrected.

**Fig. 10: EAD values on y-axis are missing**

This was added.

**p.21 l.592 "our scenarios are reasonable to represent"**

Corrected.

**Reply to Referee #2**

First of all, we would like to thank the referee for the time and effort put into reviewing the manuscript. In the following, we provide our responses to the individual questions posed by the referee.

**The objective of the paper is to contribute in filling the gap in current understanding of the roles of the different components of the risk chain on changes in flood risk. To this aim a simulation-based approach is implemented where, starting from the investigation of a baseline scenario, six components of the risk chain are changed (namely: climate change, implementation of reservoirs in the catchment, flood protection along the rivers, land use change, change in asset values and changes in the vulnerability of flood-affected objects) by both increasing and decreasing their reference value. Thanks to the implementation of the DFRA, this allows to simulate 729 damage series (of 4000 years) from which the EAD and the risk curve are derived and investigated: (i) at the catchment scale, (ii) at two typical upstream and downstream sub-basins, and (iii) for summer and winter seasons.**
**The topic of the paper is in the scope of NHESS and results can contribute to a better understanding of flood risk and risk mitigation strategies, also in the light of future climate change. The paper is overall clear, well-structured and well-written; conclusions are mostly supported by simulation results. Still, there are some points that need further clarification or that must be better explained in order to make the paper totally understandable; these are reported as "specific criticisms". On the other hand, in the following, some suggestions are reported that could increase the robustness and the completeness of the research.**

We would like to thank the reviewer for his/her positive feedback. We will consider his/her suggestions to make the paper more understandable.

**Suggestions**
**- The choice of considering a double storage capacity as the change in the catchment hydrology is not totally clear to me. In the light of choosing "plausible deviations from the baseline", it makes more sense considering a change in the operational rules like, for example, in the value of the cut-off discharge. I suggest authors to explore also this scenario;**

The reservoirs typically have multiple purposes: drinking water supply, hydropower generation and flood protection. In the baseline scenario, we implemented the reservoir volume only dedicated to the flood protection purpose. Hence, doubling of the reservoir capacity in C2 virtually means the reassigning of the operational volume for water supply/hydropower to the flood protection purpose. With this approach, it is easier to maintain the symmetry between C0 and C2 scenarios. Changing the cut-off threshold would certainly be an additional option to explore, but to maintain the symmetry is maybe not straightforward. There are many possible risk-influencing factors, but to consider all is not easy. We therefore decided to keep one changing factor for each component of the risk chain and consider changes in reservoir volume.

**Again, in the light of choosing "plausible deviations from the baseline", also the change in building quality should be investigated. This is a quite cheap strategy for risk mitigation that can be easily encouraged/achieved by public and private incentives. I suggest authors to explore also this scenario;**

Changes of building quality in respect to flood resistance via the implementation of building precautionary measures (e.g. sealing the basement, flood proofing of the building etc.) are implicitly considered in the vulnerability scenarios. Other changes of the building quality, e.g. change of building material are not easily possible or at least not as cheap and straightforward.

**The whole analysis is based on the estimation of damage to the only residential sector. Still, risk can be heavily affected by damage to other sectors like agriculture, commerce, tourism, population, etc. For some of these sectors flood damage is strongly related to the season of occurrence of the event (e.g. agriculture, tourism), and it may be the case that the effect of climate change on such sectors modifies present conclusions on EAD and on the role of the different components of the risk chain. I think that more than one sector should be included in the analysis or, at least, some considerations must be added on the possible role of damage to other exposed sectors.**

The main objective of this paper was to propose an insight to possible impacts of different components on flood risk chain under selected assumptions. In the scope of this study, we have mainly focused on changes in risk where actual (true) EAD values for the study area are not primarily targeted. We agree that the impact on other sectors might be different, but this would not refute the presented conclusions. Including additional sectors or secondary losses would further complicate the analysis and increase the volume of the manuscript destructing from the main focus which is the risk sensitivity along the process chain. Furthermore, damage modelling bears high uncertainties compared to other models along the risk chain. Loss models for other sectors than residential are even more uncertain. We therefore abstain from including damage models for other sectors into analysis.

**Specific minor comments (which can increase the readability and clarity of the paper)**
**Section 1**
**Pg. 2 line 65 "A major problem is the superposition of several drivers of risk changes" = what authors mean here with "superposition"? Please, specify**

Here, 'superposition' is standing for to explain that different drivers in the flood risk chain may mask each other's impact. Accordingly, we have given an explanation to this sentence by example in lines 66-68.

**Section 3**
**Pg. 6 line 164 "the approach provides the complete flood hydrograph" = on a daily base, is it correct?**

Yes, our approach provides the complete flood hydrograph on a daily base. This was added to the revised version of the manuscript.

**Pg. 6 line 166 "the spatial dependence between flood damages at different locations in the catchment is taken into account" = what authors mean with "spatial dependence between flood damages at different locations"? Please, specify**

With this statement we mean that damage values at different locations (e.g. subbasins) are spatially consistent though they are heterogeneous, i.e. the losses correspond to a flood/inundation footprint resulting from the continuous simulation with the risk chain model. We rephrased this sentence as follows: "Hence also spatial consistency of losses across the catchment is taken into account."

**Pg. 7 line 183 "The weather generator is parameterized on a monthly basis." = this is already stated in the following page. The sentence can be deleted**

This sentence was deleted.

**Pg. 8 lines 215 -221 = Some assumptions made for the modelling of reservoirs are not totally clear to me: (1) are reservoirs empty at the begging of the simulation? If yes, is it realistic? If no, which is the initial volume? Why? (2) What happens if the storage capacity is reached before the discharge falls below HQ100? (3) Why the return period of 100 years was chosen? Is it the design return period of dikes? (4)Which is the necessary information collected from Sächsisches Landesamt für Umwelt und Geologie?**

(1) Reservoirs can be used for different purposes such as water storage, hydropower and flood protection. Although single-purpose flood control reservoirs are rather rare, in the scope of this paper, we considered the reservoirs that have only flood control function i.e. we implemented not the total reservoir volume in the model, but only the volume dedicated for flood control. Thus, one can think that the entire reservoir is always full to its non-operation capacity or the "flood protection" reservoir volume is empty at the beginning of the simulation. Because our focus is the flood protection purpose of reservoir in the presented sensitivity analysis, we believe this is a reasonable approach.
(2) If the storage capacity was filled before the discharge falls below HQ100, excess flow is routed downstream. If this is the case, we can still expect to observe impact of the reservoir at downstream because some part of the flood volume is held by the reservoirs.
(3) The information about the operation rules were not available for this study. There is certainly a threshold for deploying the reservoirs for the flood protection purposes. Otherwise, they would be inefficient if filled passively at the time of low discharges. For this sensitivity analysis a 100 year threshold was chosen, in order to capture

large floods. Certainly, the exact risk estimates would depend on the selection of the threshold, but the general behaviour of changes in the risk curves should be invariant.

(4) We obtained information of reservoir locations and flood storage capacities from Sächsisches Landesamt für Umwelt und Geologie.

**Pg. 9 line 226 "The calibration and validation results illustrate obvious improvement in this new model setup compared to the version used in Falter et al. (2015)" = I cannot appreciate this improvement in Figure 3. On the left, only results from the new model and observations are reported so I cannot see the difference between the two models. Graphs on the right suggest that the two models are mostly equivalent. Please, comment on this**

We agree with the referee that overall the difference in model performance between the two model setups is modest looking at obtained NSE values and the plots in Figure 3. However with the statement, we would like to highlight the improvement of the new model setup with respect to the cut-off process of the extreme flood events, i.e the August 2002 flood. We modified the text accordingly.

**Pg. 9 line 231 "with the new setup, the SWIM model seems to be able to represent the cut-off process more accurately" = Of course, the old model did not consider cut-off**

We thank and agree with the referee for this comment (which has been addressed by the reviewer #1 as well). We removed the irrelevant text "more accurately" from the text.

**Pg. 10 line 254 "the minimum height was assumed at 1.8 m" = on which bases?**

Since we derived 761 cross sections of the modelled river network by extracting information from the 10 m DEM, the estimated dike height might be too low and additional dike information is not available at some locations. Therefore a threshold was introduced as a global correction value for the minimum dike height. We assumed the minimum dike height as 1.8 m following the study of Falter et al. (2015). We will explain this in the manuscript.

**Pg. 10 line 255 "the 2D raster based model uses a 100 m resampled computational grid from DEM10, which was found an acceptable compromise for representation of inundation characteristics and computation time" = I think some considerations must be included on topography. 100 m can be enough in flat areas (i.e. downstream) but can introduce big errors in damage estimation in steep areas (i.e. upstream). Did authors consider different resampling of the DEM in different areas of the catchment?**

We agree with the reviewer that 100 m resolution might be inappropriate in steep areas. Our approach is based on findings by Falter et al. (2013), which were carried out in more or less flat terrain. In this study, we focus on the sensitivity of the risk estimates rather than on estimating the actual (true) risk values in the study area. Hence, we consider the DEM resolution to be not of paramount importance for the derived conclusions, but rather a pragmatic choice to obtain plausible results from 729 simulation runs in reasonable time.

**Pg. 10 line 265 "Although there is an underestimation of inundation, the model gives a reasonable estimate of inundation extent and depth for large-scale assessments" = which are the bases for this statement? 50% underestimation in flood extent is a significant error in my point of view**

Yes, we agree with the reviewer that 50% underestimation is a significant error. We modify our statement that this is "a reasonable estimate of inundation extent" accordingly. This large error comes from the model inability to correctly represent dike breach locations and breaching processes. However, in this study we focus more on the change rather than the absolute risk value. Therefore, we decided to tolerate this mismatch and still believe this does not refute our conclusions.

**Pg. 11 lines 275-282 = I think that assumptions made for the estimation of damage must be better explained: (1) which is the scale of analysis? The 100*100 m2 cell? The municipal scale? Other? (2) how building type and level of precaution are assessed? (3) do asset values depend on building quality and type?**

We added the following notions to the manuscript to improve this section.

(1) All gridded input data (e.g. asset values and land use) were resampled to 100 m spatial resolution. The damage calculation is carried out for $100*100\ m^2$ cells and then aggregated to the level of municipalities.

(2) The composition of building types is defined using a cluster centre approach. In total five clusters are defined differentiating the share of single-family house, semi-detached/detached and multifamily houses. Average building quality is aggregated to two classes: high quality and medium/low quality (Thieken et al., 2008). Besides inundation depth, return period, building type and quality, contamination (none, medium and heavy) and private precaution (none, good and very good) are also taken into account in the damage model. The overall effect of contamination and private precaution is quantified by scaling factors for each raster cell. Building type and quality are assessed on municipality level (Thieken et al., 2008).

(3) Asset values are determined according to their reconstruction (replacement) costs. Therefore, implicitly asset values depend on building quality and type. In the damage model, the building quality and asset values are not directly related on a building-by-building basis since both characteristics are aggregated at the cell resolution.

**Pg. 11 line 283 "The sum of damages for all communities was officially reported as €240 million" = does it refer to the total damage or damage to residential buildings?**

It refers to the sum of damages to residential buildings for the August 2002 flood. The text was modified accordingly.

**Pg. 11 line 285 "This underestimation may be explained by uncertainty in asset values and their spatial distribution and uncertainty in the damage model" = and underestimation of flood extend I guess**

The reviewer is right. This underestimation is also coming from the differences in simulated and observed inundation patterns. We mentioned this in the manuscript.

**Pg. 11 line 302 "For example, the baseline scenario of the catchment component is represented by a model version calibrated for a recent time period and including the current implementation of reservoirs in the catchment. The specific time periods and assumptions for the baseline scenarios are given in sections 3.1.1 to 3.1.4 where the implementation, calibration and validation of the different modules for the current situation are described" = the meaning of the baseline scenario is clear to the reader at this point of the paper. This sentence can be omitted.**

We thank the reviewer. We omit this sentence.

**Pg. 13 line 356 – 360 = Are studies made in the Netherland transferable to the Mulde catchment? What authors mean with "potential" dike heightening? Potential with respect to what?**

We could not find any study that shows dike heightening in Germany. Therefore, to make a reasonable assumption in terms of increment range, we used examples from the Netherlands. "Potential" stands for possible increases in dike heights due to alteration in design discharge value by time. For example, in one of the dike heightening strategy, each every 5 years using additional peak discharge data, new peak discharge probability distribution is calculated and dike height is updated (Hoekstra and Kok, 2008).

**Pg. 13 line 364 "The change scenario EL2 is based on the increase in area of these two classes from 672 to 784 km2 between 1990 and 2012 where the change area was added to baseline scenario" = How the urban area was changed? I can understand this only at pg. 21**

We moved "Land use change scenarios were created based on increase in residential areas between the years 1990 and 2012 by randomly changing the state of single pixels." to pg.13 line 364.

**Pg. 13 line 367 "Pixels (100 x 100 m2) of the classes 111 and 112 were assigned to non-residential land cover classes (i.e. agricultural areas and semi-natural areas)" = how pixels were re-assigned? Why authors did not consider CORINE land use map of 1990? I think it is more realistic.**

Thanks for the reviewer for this comment. That sentence should read as follows: "Pixels (100 x 100 m2) of the classes 111 and 112 were assigned to residential land cover classes and all other classes were assigned to nonresidential land cover classes (i.e. agricultural areas and semi-natural areas)". We modified this in the revised version. It means that there is no re-assignment. The reason for not considering CORINE land use map of 1990 is the difference in representation of residential land use classes for 1990 and 2012. Therefore, to be consistent and symmetric in scenarios we created two symmetric scenarios from baseline where subtraction scenario is still realistic.

**Section 4**
**Pg. 14 line 405 "This non-symmetry in the effects of the catchment component is explained by the specific implementation of the reservoir capacity: Implementing a capacity of 106 million m3 reduces the EAD significantly, but doubling this reservoir capacity at the same locations does not further reduce the risk substantially, because the damage is primarily generated at other locations within the catchment" = not clear, the role of reservoirs is not reducing damage downstream? Please, clarify**

We agree with the reviewer that the sentence is misleading. It was rephrased as follows: "This non-symmetry in the effects of the catchment component is explained by the specific implementation of the reservoir capacity: Implementing a capacity of 106 million $m^3$ reduces the EAD significantly, but doubling this reservoir capacity at the same locations does not further reduce the risk substantially, because the reservoir capacity in the baseline scenario is already sufficient to capture floods above HQ100". We observed these by checking the cut-off volume in both scenarios and inundation extents. Differences with doubled reservoir volume were very small for most reservoirs.

**Pg. 16 line 455 "Regarding the change in catchment hydrology (C), change in flood storage capacity has a more dominant impact upstream which is explained by the reservoir locations. Due to the assumped reservoir operation the reservoir impact is only visible for very low probability events at the downstream sub-basin" = I still do not understand the influence of reservoirs in the catchment. Readers should be supported by a better description/discussion of the location of reservoirs with respect to the sub-basins.**

The size and location of the reservoirs are shown in the catchment map (Figure 1). For the analysis of the impact of reservoirs to upstream and downstream areas two specific reaches were selected (Figure 1). The (upstream) reach around Zwickau is directly downstream of a large reservoir. Doubling the capacity of this reservoir does not result in risk changes. At the downstream region influenced by several river branches, we observe aggregated impact from various reservoirs upstream. It seems that for very large events doubling of reservoir capacity still exerts a small impact on the risk downstream.

**Pg.17 line 464 "From the risk curves of different land use scenarios, it should be noted that the increased urban area scenario (EL2) increases risk upstream for high probability events" = I cannot see the difference between EL2 and the baseline scenario in Figure 7. Is one curve missing?**

There is no missing curve. In figure 7, EL2 and the baseline scenario behave almost identical upstream. It can be explained by the fact that additional upstream residential areas in EL2 scenario are not inundated. Therefore, same residential areas are inundated upstream for both baseline and EL2 scenarios.

**Pg. 17 line 468 "the baseline land use scenario (EL1) and the EL2 scenario behave almost identical upstream which can be explained by the steep topography" = I guess it depends on the rules adopted for increasing the urban area and on how the flood extent changes for different return periods**

Thanks for the comment. This is also a reason of that why we get almost identical curves upstream. We reflect on this in the manuscript.

**Pg. 17 line 472 "This can be explained by the specific setup of the residential buildings added in EL1 which are not exposed to floods." = not clear, please specify**

This is similar to the situation between baseline and EL2 scenarios upstream. In this case, the sentence implies that additional downstream residential areas in the baseline scenario compared to EL0 scenario are not inundated.

**Pg. 19 line 523 "Under the fixed A2 scenario, five scenario combinations are highlighted, each time altering a different component from its baseline value towards EAD decrease" = I can see four combinations leading to lower EAD. Could authors check?**

We removed "towards EAD decrease".

**Figures**
**Figure 1 – subcatchments are not visible in mountain areas**
We thank the reviewer. We modified Figure 1.

**Figure 2 – (1) please specify what authors mean with XS profile (2) output of the flood loss model is missing (3) level of precaution and contamination are missing in the box related to FLEMOps**

We thank the reviewer. We modified Figure 2.

**Figure 4 – I think that the figure is not explicative of the logic tree. Please, consider changes.**

We consider a change in the caption of this figure. We will change the capture of the figure. We used "conceptual scheme" instead of "logic tree".

**Figures 6, 7 and 8 are too small**
We made the fonts bigger in this figures.

**Bibliography**
**I did not check the bibliography at this stage of the review. I reserve to do this in a second time**